# Role of Gut Dysbiosis in Liver Diseases: What Have We Learned So Far?

**DOI:** 10.3390/diseases7040058

**Published:** 2019-11-12

**Authors:** Hiroshi Fukui

**Affiliations:** Department of Gastroenterology, Nara Medical University, Kashihara 634-8522, Japan; hfukui@naramed-u.ac.jp

**Keywords:** gut dysbiosis, alcoholic liver injury, non-alcoholic fatty liver disease, liver cirrhosis, liver transplantation, primary sclerosing cholangitis, primary biliary cholangitis, hepatocellular carcinoma

## Abstract

Accumulating evidence supports that gut dysbiosis may relate to various liver diseases. Alcoholics with high intestinal permeability had a decrease in the abundance of *Ruminnococcus*. Intestinal dysmotility, increased gastric pH, and altered immune responses in addition to environmental and genetic factors are likely to cause alcohol-associated gut microbial changes. Alcohol-induced dysbiosis may be associated with gut barrier dysfunction, as microbiota and their products modulate barrier function by affecting epithelial pro-inflammatory responses and mucosal repair functions. High levels of plasma endotoxin are detected in alcoholics, in moderate fatty liver to advanced cirrhosis. Decreased abundance of *Faecalibacterium prausnitzii*, an anti-inflammatory commensal, stimulating IL-10 secretion and inhibiting IL-12 and interferon-γ expression. *Proteobacteria*, *Enterobacteriaceae*, and *Escherichia* were reported to be increased in NAFLD (nonalcoholic fatty liver disease) patients. Increased abundance of fecal *Escherichia* to elevated blood alcohol levels in these patients and gut microbiota enriched in alcohol-producing bacteria produce more alcohol (alcohol hypothesis). Some undetermined pathological sequences related to gut dysbiosis may facilitate energy-producing and proinflammatory conditions for the progression of NAFLD. A shortage of autochthonous non-pathogenic bacteria and an overgrowth of potentially pathogenic bacteria are common findings in cirrhotic patients. The ratio of the amounts of beneficial autochthonous taxa (*Lachnospiraceae* + *Ruminococaceae* + *Veillonellaceae* + *Clostridiales Incertae Sedis XIV*) to those of potentially pathogenic taxa (*Enterobacteriaceae* + *Bacteroidaceae*) was low in those with early death and organ failure. Cirrhotic patients with decreased microbial diversity before liver transplantation were more likely to develop post-transplant infections and cognitive impairment related to residual dysbiosis. Patients with PSC had marked reduction of bacterial diversity. *Enterococcus* and *Lactobacillus* were increased in PSC patients (without liver cirrhosis.) Treatment-naive PBC patients were associated with altered composition and function of gut microbiota, as well as a lower level of diversity. As serum anti-gp210 antibody has been considered as an index of disease progression, relatively lower species richness and lower abundance of *Faecalibacterium spp*. in gp210-positive patients are interesting. The *dysbiosis-induced* altered bacterial metabolites such as a hepatocarcinogenesis promotor DCA, together with a leaky gut and bacterial translocation. Gut protective *Akkermansia* and butyrate-producing genera were decreased, while genera producing-lipopolysaccharide were increased in early hepatocellular carcinoma (HCC) patients.

A comprehensive PUBMED literature search including gut microbiome, dysbiosis, alcoholic liver disease, non-alcoholic fatty liver disease, liver cirrhosis, liver transplantation, primary sclerosing cholangitis, primary biliary cholangitis, and hepatocellular carcinoma was performed. This review introduced all valuable clinical research works in the world.

## 1. Gut-Liver Axis related to Gut Mycrobiomes

At first, this chapter highlights the importance of endotoxin, short chain fatty acids and bile acids because these three items are important, common bases for the entire review text. These three items surely take an important role in the relations between gut microbiome and various forms of liver injury.

## 2. Endotoxin and Other Microbial Products

In patients with liver diseases, various gut bacteria and their cell components frequently enter into the portal circulation through the disrupted intestinal barrier and reach the liver. The passage of viable bacteria from the intestinal lumen through the mesenteric lymph nodes and other sites are defined as bacterial translocation (BT). The concept of BT was later broadened to microbial products, including endotoxin, peptidoglycan and bacterial DNA [1]. 

Well-known bacterial endotoxin (i.e., lipopolysaccharide: LPS) is a component of the bacterial wall of gram-negative bacteria and acts as one of the pathogen-associated molecular patterns (PAMPs) for Toll-like receptor (TLR) 4. Translocated microbial products activate KCs in the liver through pattern recognition receptors, such as TLRs and NOD-like receptors [2]. TLRs thus recognize structural components unique to bacteria, fungi, and virus and induce innate immune responses such as cytokine production in the liver [2,3,4]. Hepatic non-immune cells, including hepatic stellate cells (HSCs) and endothelial cells, also have TLRs and respond to bacterial products [2] and proceed pro-inflammatory and profibrotic pathways through various cytokines such as IL-1, IL-6 and TNF [1].

## 3. Short Chain Fatty Acids

Human colonic microbiota breaks down substrates such as resistant starch and non-starch polysaccharides (major components of dietary fiber), which are not completely hydrolyzed by host enzymes in the small intestine [5]. The main fermentation products are short chain fatty acids (SCFAs), acetate, propionate, and butyrate [5]. Butyrate acts as an energy source for the colonic epithelium, whereas acetate and propionate serve as substrates for lipogenesis and gluconeogenesis [6,7]. The bacterial short chain fatty acids (SCFAs) thus work as additional sources of energy, constituting 3%–9% of our daily caloric intake [8]. SCFAs additionally exhibit a wide range of functions from mucoprotection, immune regulation to metabolism in variable tissues [9,10], thus having a direct and indirect effect on our bodies. The predominant bacteria that produce SCFAs are described as *Ruminococcaceae* and *Eubacterium* [10].

## 4. Bile Acids

Bile acids (BAs) are saturated, hydroxylated C-24 cyclopentanophenanthrene sterols synthesized from cholesterol in hepatocytes [11]. Cholesterol 7α -hydroxylase (CYP7A1) produces both the dihydroxy BA chenodeoxycholic acid (CDCA) and the trihydroxy BA cholic acid (CA). These primary BAs are conjugated to glycine or taurine in hepatocytes and stored in the gallbladder. Eating induces gallbladder contraction to induce emptying the contents into the small intestine [12]. Bile salts solubilize fats and fat-soluble vitamins enhancing their uptake. BAs are mostly (~95%) absorbed in the terminal ileum through the sodium-dependent BA transporter (ASBT) and are transported to the liver through the portal vein, thus forming portal enterohepatic circulation (EHC). The rest escapes the EHC and becomes substrate for microbial transformation in the right colon [11]. Conjugated primary bile acids (CDCA and CA) undergo microbial modifications (e.g., deconjugation, dehydroxylation, and hydrogenation) to form secondary bile acids lithocholic acid (LCA) and deoxycholic acid (DCA), respectively [7]. The colonic 7α-dehydroxylating bacteria (e.g., *Lachonospiraceae*, *Ruminococcaceae*, and *Blautia*) play a key role in this conversion.

BAs are recognized as signaling molecules that activate specific nuclear farnesoid X receptor (FXR) and membrane BA-activated G protein-coupled receptor (GP-BAR1) TGR5 in the intestine [13,14,15,16]. CDCA is the most potent endogenous FXR ligand. DCA and LCA, can also activate FXR. TGR5 is activated by nanomolar concentrations of LCA and micromolar concentrations of CA, DCA, and CDCA [17,18]. By activating various signaling pathways after binding to FXR in the enterocytes and the hepatocytes and to TGR5 in the non-parenchymal hepatocytes, BAs regulate various metabolic processes, such as triglyceride, cholesterol, and glucose metabolism and inflammatory reactions [18,19]. BAs on the other hand, exert negative selection pressures on gut bacteria directly through antimicrobial properties and indirectly through activation of FXR-induced anti-microbial peptide synthesis in the small intestine [19].

## 5. Gut Dysbiosis in Liver Diseases

The following sections introduce most of the outstanding research in various clinical settings that suggest an important role of gut dysbiosis in the development of various liver diseases. The sections were divided into alcoholic liver disease, non-alcoholic liver disease, liver cirrhosis, primary sclerosing cholangitis, primary biliary cholangitis and hepatocellular carcinoma. I avoided my private comments on the methodology or the main results to any investigations. Although it seems rational to show a high-level study using intestinal mucosa and the newest method, I tried to introduce all clinical studies thus far.

## 6. Alcoholic Liver Diseases

Alcoholic liver disease (ALD) is an important cause of chronic liver disease in the world and encompasses fatty liver (hepatic steatosis), alcoholic steatohepatitis, and the more severe entities such as alcoholic hepatitis, fibrosis, cirrhosis, and hepatocellular carcinoma [20,21]. In the case of persistent drinking, the fatty liver can proceed to fibrosis and cirrhosis, which can finally develop portal hypertension and liver failure [20,21]. Chronic alcohol ingestion leads to bacterial overgrowth and dysbiosis in small and large intestines in humans [21,22,23,24]. Alcohol drinking further induces changes in the composition of gut microbiota and translocation of bacterial products into the portal blood, which appears to play a cardinal role in alcohol-induced liver damage [25]. A pioneering study by Bode et al. [22] disclosed that Coliform microorganisms were cultured much more abundantly in jejunal aspirates of 27 alcoholics including 10 patients with alcoholic cirrhosis compared with that of control patients without liver disease and alcohol abuse. Subjects with long-lasting alcohol consumption, as well as patients with alcoholic liver cirrhosis were reported to present small intestinal bacterial overgrowth (SIBO) [21], a condition in which colonic bacterial translocate into the small bowel due to impaired microvilli function [26]. SIBO has been defined as ≥ 10^5^ total colony-forming units per milliliter of proximal jejunal aspirations. At the same time, plasma endotoxin levels are elevated in chronic alcoholics from the early stages of ALD [27]. SIBO and gut dysbiosis are likely to provoke increased intestinal permeability, pathological BT and endotoxemia. All of these are considered to impact the progression of ALD.

To date, there are several manuscripts published on the effect of alcohol drinking on the fecal microbiome. The experimental data obtained are not constant and probably depend on the different methods of alcohol feeding. The intragastric feeding of alcohol (Tsukamoto-French method) to mice for three weeks was reported to induce a relative decrease of *Firmicutes* (*Lactococcus*, *Pediococcus*, *Lactobacillus*, and the *Leuconostoc* genera) and a relative increase of *Verrucomicrobia* and *Bacteroidetes* (*Bacteroidales*, *Bacteroides*, and *Porphyromonadaceae*) [28]. In an experimental model in mice with the chronic ad libitum ethanol feeding of the Lieber-DeCarli ethanol liquid diet for eight weeks, fecal *Bacteriodetes* was conversely decreased together with *Firmicutes,* while *Proteobacteria* and *Actinobacteria* were increased [29]. The latter study further proved the biggest expansion of gram-negative alkaline-tolerant *Alcaligenes* and gram-positive *Corynebacterium* [29]. Another study indicated that the administration of ethanol in the drinking water for seven days to mice increased *Enterobacteriaceae* in the contents of the small intestines [30]. *Proteobacteria* includes several pathogenic species such as *Salmonella*, *Helicobacter*, *Vibrio* and *Escherichia*. The [31] *Enterobacteriaceae* group including *Escherichia coli* (*E. coli*) can be related to several pathological conditions [30].

Changes in intestinal microbiota reported in clinical studies in patients with ALD are summarized in Table 1. Kirpich et al. [32] compared the fecal microbiota of 66 alcoholics with 24 healthy controls and found reduced number of cultured *Bifidobacteria*, *Lactobacilli*, and *Enterococci* in alcoholic patients compared with control subjects. Mutlu et al. [33] analyzed colonic biopsy samples by the 16S rRNA gene pyrosequencing and found that the mean abundance of *Bacteroidaceae* in *Bacteroidetes* was decreased in alcoholics compared with healthy controls. Their study further reported that alcoholics with dysbiosis (11 of 41 patients) had lower abundances of *Bacteroidetes* and *Clostridium* and higher abundances of *Proteobacteria, Bacilli* and γ-*Proteobacteria* compared with alcoholics without dysbiosis (30 of 41 patients) [33]. The duration of sobriety was not related to the presence of dysbiosis in their sober alcoholics, which indicated that the effects of chronic alcohol drinking on microbiota were long-lasting [33].

Leclercq et al. [34] found that 26 of 60 liver fibrosis stage F0~F1 patients with alcohol dependence had increased intestinal permeability (IP). Interestingly enough, only patients with increased IP exhibited an altered fecal microbiota composition. In these patients, *Ruminococcaceae* and *Incertae Sedis XIII* were less abundant using culture-independent methods, whereas *Lachnospiraceae* and *Incertae Sedis XIV* were more abundant compared with alcohol-dependent subjects with low IP and controls [34]. At the genus level, alcoholics with high IP had a marked decrease in the abundance of *Ruminococcus*, *Faecalibacterium*, *Subdoligranulum*, *Oscillibacter*, and *Anaerofilum* belonging to the *Ruminococcaceae* family. The abundance of *Dorea* belonging to the *Lachnospiraceae* family increased in alcoholics with high IP [34]. Additionally, the genera *Blautia* and *Megasphaera* were increased whereas *Clostridium* was decreased in alcohol-dependent subjects with high IP [34]. Their analysis further revealed that the total amount of bacteria and those belonging to the *Ruminococcaceae* family, especially *Faecalibacterium prausnitzii* (*F. prausnitzii*), were negatively correlated to IP, while the genera *Dorea* and *Blautia* were positively correlated with IP. A butyrate-producing anti-inflammatory commensal *F. prausnitzii* was further negatively correlated with plasma Interleukin (IL)-8 levels [34]. These findings support their conclusion that alterations in microbial composition are under the influence of increased IP and proinflammatory cytokine responses [35]. Intestinal SCFAs are decreased after alcohol drinking except for acetic acid, which conversely increases as a metabolite of ethanol [21,36]. In addition to *F. prausnitzii*, two butyrate-producing bacteria *Ruminococcus* [34] and *Coprococcus* [37] were reported to be decreased in the feces of alcoholics. As explained above, butyrate is a cardinal source of energy for enterocytes and influences the intestinal barrier function through the stimulation of tight junctions and mucous production.

Compared with HBV-related cirrhotic patients, Chen et al. [38] reported more enriched fecal *Prevotellaceae* in Chinese patients with alcoholic cirrhosis using pyrosequencing of the 16S ribosomal RNA V3 region. They discussed that increased *Prevotellaceae* may be related to ethanol metabolism in human gut [38]. On the other hand, Kakiyama et al. [39] found a decrease in *Bacteroidaceae* and *Porphyromonadaceae,* and an increase in *Veillonellaceae,* in American drinkers compared with nonalcoholic cirrhosis. In this report, *Veillonellaceae* is specifically linked with higher systemic inflammation and endotoxemia in cirrhotics [39].

Bajaj et al. [40] further reported that American patients with alcoholic cirrhosis had a higher abundance of *Enterobacteriaceae* and *Halomonadaceae,* and a lower abundance of *Lachnospiraceae, Ruminococcaceae,* and *Clostridialies XIV* despite a similar MELD score compared with those without alcoholic etiology. They thought these dysbiosis may partly explain the higher inflammatory state and bacterial translocation in patients with alcoholic cirrhosis [40]. Continuous alcohol drinking predisposes cirrhotic patients to evident gut dysbiosis, i.e., decreased autochthonous taxa (*Lachnospiraceae, Ruminococcaeae*, and *Clostridiales cluster XIV*) in the intestinal mucosa [41]. In the ascending colon, an increased proportion of *Proteobacteria* (*Enterobacteriaceae*) and decreased abundance of *Lachnospiraceae, Bacteroidaceae*, and *Prevotellaceae* were seen in alcoholic cirrhosis compared with non-alcoholic cirrhosis [41].

There is no clear explanation for finding that the two autochthonous taxa, *Lachnospiraceae* and *Clostridiales cluster XIV*, were rather increased in the feces of alcoholics with mild liver fibrosis and increased IP in the above study by Leclercq et al. [34]. It may be possible that these taxa were increased to counteract against leaky gut and intestinal inflammation in an early stage of alcoholic liver disease. They may be finally decreased in advanced alcoholic cirrhosis as shown by Bajaj et al. [40,41].

Various factors including intestinal dysmotility, increased gastric pH, and altered immune responses in addition to environmental and genetic factors might contribute to alcohol-associated microbial changes [21]. In addition, various antimicrobial proteins, produced in the intestinal epithelium and Paneth cells, may also impact on gut microbiome with their abilities to kill or inactivate microorganisms [42]. Bacterial recognition by TLRs on intestinal epithelial cells is required for the expression of antimicrobial Reg3β and Reg3γ [42]. Alcohol feeding induces down-regulation in gene and protein expression of Reg3β and Reg3γ in the mucosa of the small intestine, which may contribute to the changes in intestinal microbiome [28]. MyD88-dependent Reg3γ exhibits a potent bactericidal activity in intestinal epithelial and Paneth cells [43,44]. TLR2-dependent Reg3β in Peyer’s patches is an essential factor in conditioning epithelial defense signaling pathways against bacterial invasion [45].

The lowest levels of Reg3β and Reg3γ were noted in the proximal small intestine of mice, where the bacterial overgrowth is prominent and luminal alcohol concentrations are highest [28]. Reg3γ^−/−^ and Reg3β^−/−^ mice have increased susceptibility to ethanol-induced liver injury in relation to increased mucosal bacteria and BT, whereas intestine-specific overexpression of Reg3γ protects mice against it [42,46].

As described above, Mutlu et al. [33] found a lower abundance of *Bacteroidetes* and higher abundance of *Proteobacteria* in the colonic mucosal microbiome of alcoholics. The observed dysbiosis appeared to be parallel with high levels of serum endotoxin [33]. 

Endotoxemia and hepatic inflammation are regarded as consequences of the expansion of the gram-negative bacteria from the *Proteobacteria* phylum, which occur in response to chronic ethanol drinking. A decrease in commensal bacteria could contribute to defecting the protective tight junction barrier [47]. The probiotic *Lactobacillus rhamnosus GG* suppressed the fecal dysbiosis (the overgrowth of *Alicaligene* and *Corynebacterium*), improved the decrease in tight junction protein expression, and attenuated endotoxemia and related hepatic injury in alcohol-fed mice [29]. These findings may provide a rationale for the beneficial effect of various *Lactobacillus* strains [28]. The increases of *Lactobacillus spp.*, *Bifidobacterium spp.* and the family *Ruminococcaceae* during alcohol abstinence suggested that these bacteria have a beneficial impact on gut-barrier function and could contribute to the maintenance of IP [35]. Although the early culture study by Kippich et al. [32] reported decreased *Lactobacillus* and *Bifidobacterium,* the following non-culture methods could not confirm these changes. Conversely, increased fecal abundance of *Bifidobacterium* were observed in patients with severe alcoholic hepatitis [48] and alcoholic liver cirrhosis [37]. Certain species of *Lactobacillus* were enriched in the feces of patients with alcohol dependence and alcoholic cirrhosis [37]. Lactobacillus species are profitable bacteria, which produce bacteriocins, like antibiotics [49]. As the clinical effects of *Lactobacillus spp.* and *Bifidobacterium spp.* are theoretically based on the decreases of these bacteria in stool of patients, further elaborate metagenomic analysis of fecal microbiome is necessary related to the usefulness of probiotic therapy.

Recently, Yang et al. [50] reported that alcohol-dependent patients displayed reduced intestinal fungal diversity and a dramatic overgrowth of *Candida*, with concomitant decreases in *Epicoccum*, *Galactomyces* and *Debaryomyces*. Patients with alcoholic cirrhosis seemed to have increased systemic exposure and immune response to mycobiota compared with healthy subjects and patients with non-alcoholic cirrhosis [50]. Moreover, the levels of extraintestinal fungal exposure estimated by serum anti–*Saccharomyces cerevisiae* IgG antibodies correlated with mortality [50], which suggest intestinal micobiome may affect the clinical course of alcoholic cirrhosis in addition to gut microbiome.

### 6.1. Intestinal Hyperpermeability and Bacterial Translocation

In a study on the intestinal permeability by way of a 51Cr-EDTA absorption test [51], non-cirrhotic alcoholic patients abstaining from alcohol for less than four days invariably showed higher intestinal permeability than controls, and the abnormality persisted for up to two weeks after cessation of drinking in many patients. In another study by a sugar test such as urinary lactulose/mannitol ratio [52], alcoholics with chronic liver disease exhibited a marked and significant increase in intestinal permeability, although those without liver injury had also a small increase in permeability. The liver receives portal blood containing microbial products and acts as the initial site of their filtration and detoxication [1]. BT is initiated when the intestinal epithelium is destructed and the intestine becomes more permeable [21,53]. Alcohol is known to disrupt the gut epithelial barrier integrity [54], resulting in the translocation of potentially harmful bacteria and their products such as endotoxins [22,55]. The intestinal dysbiosis and dysregulated bile acid metabolism in alcohol drinking might trigger tight junction dysfunction by inducing subclinical intestinal inflammation [49].

Alcohol and its metabolites, acetaldehyde and fatty acid ethyl esters, may contribute to the disruption of tight junctions, mainly through nitric oxide-mediated oxidative tissue damage and alterations in the cytoskeleton, but also through direct cell damage [56,57].

Alcohol-induced intestinal dysbiosis may be associated with intestinal epithelial barrier dysfunction [17,58] as microbiota and their products modulate barrier function by affecting epithelial pro-inflammatory responses, protein expression, and mucosal repair functions [58]. Dysbiosis-induced intestinal inflammation and tumor necrosis factor (TNF) α receptor I signaling in intestinal epithelial cells were proved to mediate a disruption of the intestinal barrier in alcohol-fed mice [59].

### 6.2. Endotoxemia and Its Consequences

Fukui et al. [27] found high levels of plasma endotoxin in alcoholics, not only in those with advanced cirrhosis but also in those with moderate fatty liver, suggesting alcohol abuse causes endotoxemia irrespective of the severity of liver disease. There are strong evidences supporting that gut-derived bacterial endotoxin plays a key role in the initiation and progression of alcoholic liver injury and the development of organ failure in some cases [33,60]. Once it reaches various organs, it strongly stimulates TLR4 in macrophages (e.g., Kupffer cells in the liver), which activates downstream signaling activation of the nuclear factor-kB pathway and induction of proinflammatory cytokines (e.g., TNF-α, IL-6, IL-8) and chemokines [61]. In addition, it indicates that gut microflora-derived endotoxin and TLR4 contribute to the progression of liver fibrosis [4]. Alcohol induces LPS binding protein (LBP) and TLR4, thus enhancing responsiveness to endotoxin. The binding of LPS to CD14/TLR4 on Kupffer cells (KCs) stimulates production of cytokines, chemokines and reactive oxygen species, which leads to T lymphocyte and neutrocyte recruitment, hepatic stellate cell (HSC) activation, and collagen production in patients with alcoholic steatohepatitis [1,3]. It is important that hepatic non-immune cells, such as HSC and endothelial cells, also respond to bacterial products through TLRs [2].

The pathophysiological role of endotoxemia is further essential in patients with severe alcoholic liver injury. Plasma endotoxin levels were elevated with the progression of alcoholic liver injury and reached the maximal level in patients with alcoholic cirrhosis and severe alcoholic hepatitis who exhibited marked hypercytokinemia [62].

### 6.3. Dysbiosis and Metabolic Changes

Bajaj et al. [41] reported that stool metabolomics showed widespread alteration in active cirrhotic drinkers. On the analysis by GC/MS, metabolites focused on bioenergetics (citrate, malate, phosphate) and amino acids (threonine, ornithine, serine) were significantly depressed in actively drinking cirrhotic patients compared with those of abstinent cirrhotic patients [41]. The decrease in these metabolites might be due to increased consumption by the gut microbiota, because they are shared between the host and the microbiota [41]. An experimental study reported that chronic ethanol administration results in a decrease in SCFAs except for acetic acid, a reduction of almost all amino acids including three branched-chain amino acids, perturbations of the steroid, lipid and carnitine metabolism in the rat GI tract [36]. Cirrhotic patients with active alcohol drinking present higher total bile acids, elevated secondary bile acids (lithocholic acid and deoxycholic acid) in duodenal fluid and stool, and an enriched glycine-conjugated BAs in duodenal fluid [21,39,42]. The shift of bile acids profile toward potentially toxic secondary bile acids is related to enhanced expression of inflammatory cytokines in colonic mucosa [39], and may lead to intestinal and systemic inflammation together with widespread dysbiosis [41]. BAs-related receptors such as FXRα, FGF-19, ASBT, and SHP in the terminal ileum of these patients were appropriately activated in response to the augmented BA load [41]. It is unclear why these reactions failed to suppress the continuous BA secretion and its enterohepatic cycling in this situation. On the other hand, serum FGF19 and bile acids were reported to be increased in patients with alcoholic hepatitis, where de novo bile acid synthesis marker (*CYP7A1* gene expression and C4 serum levels) were decreased [63]. The same study group reported FXR activity and FGF15 protein secretion were rather depressed despite increased intestinal unconjugated BA in mice with chronic ethanol feeding [64]. In this experiment, targeted interventions to improve bile acid-FXR-FGF15 signaling reduced ethanol-induced liver injury by modulation of hepatic Cyp7a1 [64]. Taken together, the discrepancies between these clinical and experimental studies should be appropriately fixed. The literature thus far is conflicting regarding efficacy. It is tempting that this BA dysregulation related to gut dysbiosis could become an innovative therapeutic target for alcoholic liver disease, however it deserves future studies.

## 7. Nonalcoholic Steatohepatitis (NASH)

Nonalcoholic fatty liver disease (NAFLD) is the hepatic manifestation of the metabolic syndrome. It includes a spectrum of pathological changes ranging from the simple steatosis (nonalcoholic fatty liver: NAFL) in the liver through nonalcoholic steatohepatitis (NASH) to fibrosis, cirrhosis, and finally hepatocellular carcinoma [65]. In addition, data show that NAFLD correlates with increased cardiovascular risk and most of the clinical surrogates of cardiovascular diseases [66]. It is important to bear in mind that an increased consumption of fructose may result in an increased lipid accumulation in the liver, which was related to insulin resistance and increased plasma triglycerides [67]. There is strong evidence for the relation of gut microbiota and bacterial endotoxin in the mechanisms of hepatic steatosis and its progression to NASH.

### 7.1. Obesity, NASH and Dysbiosis

Bacterial overgrowth, immune dysfunction, alteration of the luminal factors, and altered intestinal permeability are all recognized to be implicated in the pathophysiology of NASH and its complications [67]. Most controlled trials using breath tests demonstrated higher prevalence of SIBO (50%~77.8% vs. 9.1%~31.2%) in NAFLD patients compared with healthy subjects [65,68,69,70,71,72]. In contrast to these data, total bacterial counts in the feces estimated by real-time PCR, did not differ between healthy controls and patients with NAFLD or NASH [73]. Further studies are needed to determine whether fecal bacterial counts actually reflect the amounts of microbes present in the small intestine [74].

As for changes in fecal microbial communities, the first preliminary study reported a higher abundance of *Firmicutes* and lower abundance of *Bacteroidetes* in obese individuals compared with lean controls [75]. There has been accumulating experimental and clinical evidence on the close relation of gut microbiota to obesity. In obese mice, it has been established that the proportion of *Firmicutes* is increased and that of *Bacteroidetes* is decreased, therefore the *Firmicutes/Bacteroidetes* ratio is increased relative to their lean controls [76]. In a metagenomic and biochemical analysis in *ob/ob* obese mice [77], these microbial changes affected the metabolic state of the mouse gut microbiota in the way that their microbiome has an increased capacity to harvest energy from the diet. In this study, lean germ-free recipients get greater fat deposition by the fecal microbial transplantation from high-fat diet-fed obese mice than recipients from lean donors [78]. Hildebrandt et al. [79] studied the microbiome of wild-type and resistin-like molecule (RELM) β knockout (KO) mice switching from a standard chow diet to a high-fat diet. Although both genotypes showed extensive changes in gut microbiome including a decrease in *Bacteroidetes* and an increase in both *Firmicutes* and *Proteobacteria*, RELMβ KO mice remain comparatively lean on a high-fat diet, indicating that the high-fat diet and not the obese state, predisposes to the dysbiosis [79]. Turnbaugh et al. [78] transplanted human fecal microbial communities into germ-free mice and found that the high-fat, high-sugar “Western” diet shifted the structure of the microbiota and changed the representation of metabolic pathways in the microbiome. These mice had increased adiposity and their microbiota revealed increased *Erysipelotrichi* and *Bacilli* (mainly *Enterococcus*) within the *Firmicutes,* as well as a decreased representation of the *Bacteroidetes* members [78].

Ley et al. [75] considered that increased *Firmicutes* and decreased *Bacteroidetes* is a typical pattern of human obesity. [75]. The greater *Firmicutes/Bacteroidetes* ratio and increased *Proteobacteria* shows negative health effects, including increased energy harvest from the diet, as well as increased systemic inflammation by enhancing gut permeability [80]. However, the clinical studies on fecal microbiome have given variable and sometimes contradictory results [8,81,82,83,84,85,86,87]. Several studies [8,84,85] even reported a decreased *Firmicutes/Bacteridetes* ratio in obese subjects, although they sometimes showed increased abundance of *Proteobacteria* [84,85].

Changes in intestinal microbiota of patients with NAFLD and NASH are summarized in Table 2. Wong et al. [88] reported that Chinese NASH patients had lower fecal abundance of *Faecalibacterium* and *Anaerosporobacter* and higher abundance of *Parabacteroides* and *Allisonella*. The order *Aeromonadales*, the families *Succinivibrionaceae* and *Porphyromonadaceae*, and the genera *Parabacteroides* and *Allisonella* were more abundant in NASH patients than controls [88]. On the other hand, the class *Clostridia*, the order *Clostridiales*, and the genera *Faecalibacterium* and *Anaerosporobacter* were less abundant in NASH patients [88]. Decreased abundance of *Faecalibacterium* was also reported in a recent Canadian study [89], which also exhibited a lower abundance in *F. prausnitzii* in both NASH and simple steatosis patients compared with healthy control. As written in the section of alcoholic liver disease, *F. prausnitzii* is an anti-inflammatory commensal, which stimulates IL-10 secretion and inhibits IL-12 and interferon-γ expression [35]. The abundance of *Firmicutes* was reported to be lower in Canadian NAFLD [89] patients and American and Chinese NASH patients [84,88] compared with healthy controls. On the other hand, another Canadian study [90] showed an over-representation of selected members of phylum *Firmicutes* (*Lachnospiraceae*, *Dorea*, *Robinsoniella*, *Roseburia* and *Lactobacillus* species), while *Ruminococcaceae*, *Porphyromonadaceae*, and *Oscillibacter* were under-represented in the feces of NAFLD patients. Zhu et al. [84] reported that *Proteobacteria*, *Enterobacteriaceae*, and *Escherichia* were the only phylum, family, and genus exhibiting significantly increased levels in fecal microbiomes of young American NASH patients compared with obese children. They related increased abundance of fecal *Escherichia* to elevated levels of blood alcohol in these patients and proposed an attractive alcohol hypothesis that gut microbiota enriched in alcohol-producing bacteria (e.g., *E. coli*) produce more alcohol than healthy microbiota and supply with reactive oxygen species to the liver [84]. Other Chinese study groups also reported enriched *Proteobacteria* [91], *Enterobacteriaceae* [91] and *Esherichia* [91,92], in NAFLD patients. Endogenous alcohol produced by bacteria (e.g., *E. coli*) is further assumed to increase gut permeability and to promote inflammation in the liver [92].

Mouzaki et al. [73] once reported that adult Canadian patients with NASH had a lower percentage of *Bacteroidetes* in their feces compared with both simple steatosis and healthy controls. However, their recent qPCR study on the selected fecal taxa [89] failed to prove its reduction in NAFLD patients. Another recent study [93] showed reduced *Bacteroidetes* and increased *Actinobacteria* in Italian NAFLD children compared with healthy controls. This study additionally reported increased levels of *Bradyrhizobium*, *Anaerococcus*, *Peptoniphilus*, *Propionibacterium acnes*, *Dorea*, and *Ruminococcus* and reduced proportions of *Oscillospira* and *Rikenellaceae* compared with control subjects [93]. Marked differences between their results and other results may reflect variations in environmental and dietary factors, as well as age, among the patient cohorts [93]. Coming back to the fecal levels of *Bacteroidetes*, these studies are in quite contrast to those by Zhu et al. [84] and Sobhonslidsuk et al. [94], which showed enriched *Bacteroidetes* in NASH patients compared with healthy controls. Although the reason for this discrepancy is far from clear, marked increase in *Prevotella* within *Bacteroidetes* phylum is the common finding in the latter two studies [84,94]. Sobhonslidsuk et al. [94] discussed advanced fibrosis stages in their patients compared with those in other studies may explain the increased fecal abundance of *Bacteroidetes*, as Boursier et al. [95] reported higher fecal abundance of *Bacteroidetes* in NAFLD patients with significant fibrosis than those with mild fibrosis.

Among various factors, dietary habit is considered to be most influential on gut microbiome in obese individuals and NAFLD patients. It is well known that high-fat Western style diet causes gut dysbiosis characterized by lowered species richness (α- diversity) and changes in microbial composition, such as decreased *Bacteroidetes* and increased *Firmicutes* and *Proteobacteria* [80]. Healthy European children eating a Western diet were reported to exhibit more abundant fecal *Firmicutes* and *Proteobacteria* (especially LPS-releasing *Enterobacteriaceae*), less abundant fecal *Actinobacteria* and *Bacteroidetes*, and less fecal SCFAs contents compared with a rural African child eating high-fiber low-fat diet [96]. The feature of gut dysbiosis is especially variable in patients with NAFLD. *Prevotella* is associated with plant-rich diets as a member of phylum *Bacteroidetes*.

*Prevotella* dominated microbiota has higher fiber utilizing capacity compared with *Bacteroides* dominated microbiota, producing higher amount of SCFAs (2–3 times more propionate) [97]. It has been thus considered as beneficial commensal bacteria. Two Chinese studies [91,92] reported a decreased abundance of *Prevotella* in the feces of NAFLD patients compared with healthy subjects. However, Zhu et al. [84] reported increased fecal *Prevotella* abundance in American obese or NASH children compared with lean healthy children.

Sobhonslidsuk et al. [94] also noted enriched fecal *Prevotella* in NASH patients in Thailand. The increase in fecal *Prevotella* was further reported in Chinese cirrhotic patients [98]. These contradictory results may be partly explained by the differences in dietary factors, age or stages between the studies. However, it should be noted that emerging human studies have linked the increased abundance of fecal *Prevotella* species to intestinal and systemic inflammatory diseases. Except for abundant healthy strains, the genus *Prevotella* may include certain pathogenic strains that thrive in an inflammatory environment and stimulate Th17- mediated inflammation [99]. In this sense, future studies on *Prevotella* should be directed to characterize properties at the species level and to evaluate these species in different disease stages.

### 7.2. Dysbiosis and Progression of NAFLD

Shen et al. [91] reported that Chinese NAFLD patients with moderate fibrosis (F≥2) had a higher abundance of genus *Escherichia*, *Shigella* and the corresponding *Enterobacteriaceae* family than those with F0/F1 mild fibrosis. Özkul et al. [100] found increased *Enterobacteriaceae* and decreased *Akkermansia muciniphila* (*A. muciniphila*) in their Turkish NASH patients and reported that those with moderate F ≥ 2 fibrosis also had a higher abundance of *Enterobacteriaceae* than those with F0/F1 fibrosis. A low abundant mucous bacteria *A. muciniphila* is known to elevate the intestinal endocannabinoids levels and to control inflammation, increase gut barrier and peptide secretion [101]. This microbiome is also known to reverse high-fat diet-induced metabolic disorders, such as fat-mass gain, endotoxemia, adipose tissue inflammation, and insulin resistance [101].

Boursier et al. [95] further reported that their NAFLD patients with significant fibrosis (F ≥ 2) had a higher abundance of fecal *Bacteroides* than those with mild F0/F1 fibrosis in France. Their NASH patients showed greater abundance of *Bacteroides* and *Ruminococcus* and smaller amount of *Prevotella* compared with non-NASH patients [95]. Increased *Bacteroides* and decreased *Prevotella* in the feces of NASH patients is in line with above- mentioned information on the relationship between diet and gut microbiome [95] and may be regarded as the proinflammatory gut dysbiosis with the progression of NAFLD. The authors explained the effect of increased *Ruminococcus* by a possible increase in deleterious proinflammatory species (e.g., *R. gnavus*) within the genus, on the bases that the *Ruminococcus* genus is quite heterogeneous [95]. In fact, reclassification of some proinflammatory species originally classified to *Ruminococcus* has been discussed [102].

Furthermore, Loomba et al. [103] characterized the gut microbiome compositions using whole genome shotgun sequencing of DNA extracted from stool samples of 86 patients with biopsy-proven NAFLD and reported that *Firmicutes* is higher in mild/moderate NAFLD (stage 0–2 fibrosis) while *Proteobacteria* was higher in advanced fibrosis (stage 3 or 4 fibrosis). At the species level, the abundances of *Ruminococcus obeum* and *Eubacterium rectale* were significantly lower in advanced cases. They also found a trend of increase in *E. coli* in advanced fibrosis and demonstrated that the dysbiosis including *E. coli* dominance occurs earlier in the stage of fibrosis and may precede development of portal hypertension [103].

Although reported earlier in this review, there have been various reports in patients with NAFLD, it has been generally admitted that decrease in anti-inflammatory authochtonous taxa and increase in proinflammatory pathological taxa. In any case, the effect of diet for gut microbiome of these patients seems great.

We should always consider the effect of diet on gut microbiome before evaluating the impact of dysbiosis on the pathogenesis of NAFLD, as the disease is closely related to unhealthy dietary habit. Prolonged Western style diet causes increases in gram-negative bacillus in the gut promoting increased intestinal permeability and greater LPS/endotoxin delivery into portal and systemic circulation, a condition known as metabolic endotoxemia [80].

Salzman et al. [104] thought that the hepatotoxic consequences of intestinal dysbiosis are mixed with intestinal microbiota-mediated inflammation of the local mucosa that encourages mucosal immune dysfunction, thus presenting important plausible insight in NAFLD pathogenesis.

### 7.3. Metabolic Changes Related to Gut Microbiota

In addition to microbial cells or microbial structural components, microbial metabolites also have the ability to affect the health and disease of the host. Contradictory findings on the bacterial taxonomic composition of patients in relation to the development of NAFLD may support the importance of metagenomic analysis combined with metabolomics [74]. However, this kind of study is still scarce.

Obesity is accompanied by an intestinal dysbiosis that has an increased capacity to collect energy from the diet. Increased production of SCFAs by the gut microbiota was first reported in *ob/ob* mice [77]. Schwiertz et al. [8] found an increased total amount of fecal SCFA and propionate in obese subjects, which is in line with the thought that SCFA metabolism may play an important role in the progression of obesity. In contrast to the previous reports with regard to the contribution of bacterial groups [75,81,82], the fecal *Firmicutes/Bacteridetes* ratio was decreased in favor of the *Bacteroidetes* in obese subjects in their study [8]. The authors discussed that the ratio of *Firmicutes* and *Bacteroidetes* may not be so important compared with the amount of produced SCFA for weight control [8], as other studies reported that diets designed to achieve weight loss in obese subjects decreased fecal SCFA without any change in this ratio [105,106]. G protein-coupled receptor 43 (GPR43), a receptor for SCFAs, controls inflammatory responses in the gut [107]. GPR43, also termed free fatty acid receptor 2 (Ffar2), is expressed in the colonic mucosal cells and stimulates peptide YY release from neuroendocrine L cells, which slows gastric emptying and intestinal transit hence enhancing nutrient absorption. These L cells also release glucagon-like peptide 1 (GLP-1), which increases glucose-dependent insulin secretion. SCFAs/GPR43 signaling is recognized to improve gut permeability and minimize the hepatic injuries imposed by microbial cell components and products [107]. SCFAs, especially butyrate, may also suppress inflammation via effects on T regulatory cells in the mucosa 6, [72,73]. On the other hand, GPR43 present in intestinal neutrophils is considered to increase intestinal inflammation and permeability [108]. The fact that GPR43-deficient (Ffar2-KO) mice are completely protected from high-fat diet-induced obesity, dyslipidemia, and fatty liver [109] supports the importance of SCFAs/GPR43 signaling in the development of NAFLD. These contradictory actions of SCFAs/GPR43 signaling may cause its complex effect on the progression of NAFLD. The balance of different SCFAs produced in the gut under the variable dietary, microbial and inflammatory conditions might determine their net effect on intestinal inflammation, permeability as well as metabolism in the body [6] (Figure 1).

*SCFAs/GPR43* signaling is recognized to improve gut permeability and minimize the hepatic injuries imposed by microbial cell components and products. SCFAs, especially butyrate, may also suppress inflammation via effects on T regulatory cells in the mucosa. On the contrary, GPR43 present in intestinal neutrophils is considered to increase intestinal permeability and inflammation. These contradictory actions of SCFAs/GPR43 signaling may cause its complex effect on the progression of NAFLD.

Several researchers [84,90,92] reported deceased abundance of SCFA-producing *Ruminococcaceae* in the feces of obese NAFLD and NASH patients, which contradicts the previous results showing enriched fecal SCFAs [8,105]. Further studies should focus on gut microbiome and SCFA levels by longitudinal trials in large cohorts. It is plausible that some undetermined pathological sequences related to gut dysbiosis may facilitate energy- producing and proinflammatory condition for the progression of NAFLD.

Ferslew et al. [110] found that patients with NASH had significantly higher fasting and post-prandial serum total BAs levels as well as taurine- and glycine-conjugated primary and secondary BAs levels compared with healthy subjects. Mouzaki et al. [111] reported that patients with NAFLD, particularly those with NASH, had higher total fecal bile acid levels and enhanced bile acid synthesis in the liver. The secondary to primary BA ratio in the stool was lower in NASH compared with healthy control, but ratio of conjugated to unconjugated BAs was not different between the groups [111]. Fecal *Bacteroidetes* and *Clostridium leptum* (*C. leptum*) were decreased in patients with NASH [111]. As *C. leptum* is capable of converting primary BAs to secondary BAs by performing 7α-dehydroxylation and deconjugation, they discussed lower ratio of secondary to primary BA could be attributable to decreased *C. leptum* [111]. They further found that primary unconjugated fecal BAs correlated with the degree of hepatic steatosis, the presence of ballooning, and the severity of fibrosis. This finding together with increased serum BAs [110] suggests the hepatotoxic impact of hydrophobic BAs including stellate cell activation and initiation of cell necrosis [111,112]. It is a mystery why the increased intestinal BAs failed to suppress the hepatic de novo BA synthesis here again. Serum FGF19 levels was reported to be reduced [113,114] or unchanged [111,115] in patients with NAFLD. There might be some impairment in the regulation of bile acid synthesis just like as in ALD.

Jiao et al. [116] recently reported that increased hepatic gene expression of BA synthesis (7α-hydroxylase, Sterol 27-hydroxylase and 12α-hydroxylase) and transport protein (Na^+^- taurocholate co-transporting polypeptide, organic anion transport protein B1, organic anion transport protein B3 and bile salt export protein) together with disproportionally increased serum DCA and decreased serum FGF19 levels in NASH patients. The metagenomic analyses with functional profiling of microbiome revealed elevations of two BA-related pathways ‘glycine, serine and threonine metabolism’ and ‘taurine and hypotaurine metabolism’ and a distinct enrichment of taurine- and glycine-metabolizing bacteria *Esherichia, Bilophila and Rhodbacter* in NASH fecal samples [116]. These taxa are important for deconjugation of primary BAs prior to the downstream metabolism to secondary BA [116]. Although the authors speculated that increased FXR antagonist DCA may suppress FXR signaling in the liver and the gut [116], future comprehensive study should be directed to the functional analysis of microbial BA converting enzymes as well as other related BAs in the intestine. Proposed relationships between gut microbiome, BAs and NAFLD/NASH are shown in Figure 2.

BAs and NAFLD/NASH. BAs are essential for the maintenance of normal gut-microbiome-liver axis. They regulate gut microbiome, preventing their overgrowth and BT as described later in the section of liver cirrhosis. Beneficial effects of BAs and their agonists such as FXR and TGR5 ligands on lipid and glucose metabolism and liver pathology have been reported. The secondary to primary BA ratio in the stool was lower in NASH compared with healthy control, but ratio of conjugated to unconjugated BAs was not different between the groups. It is a mystery why the increased intestinal BAs failed to suppress the hepatic de novo BA synthesis. Serum FGF19 levels were reported to be reduced or unchanged in patients with NAFLD. There might be some impairment in the regulation of bile acid synthesis just like as in ALD.

Of note, BAs are essential for the maintenance of normal gut-microbiome-liver axis. They regulate gut microbiome, preventing their overgrowth and BT as described later in the section of liver cirrhosis. Beneficial effects of BAs and their agonists such as FXR and TGR5 ligands on lipid and glucose metabolism and liver pathology have been reported one after another [117,118,119,120,121,122,123,124,125,126]. It is known that FXR regulates gluconeogenesis via phosphoenolpyruvate carboxykinase, which ameliorates lipid and glucose metabolism and prevents inflammation [6]. Evolving new strategies for the management of fatty liver and associated metabolic disorders along these lines has attracted research interest, before exploring the relationships of these BAs receptors to gut microbiota and the metabolic syndrome. In addition, the beneficial effect of FXR ligands on liver inflammation and fibrosis [118,119,120,121], the fact that FXR-deficient mice fed a methionine/choline-deficient diet (MCDD) developed more severe liver injury but a lower degree of steatosis suggests the role of BAs and FXR in maintaining liver homeostasis against metabolic syndrome [127].

The effect of the gut microbiome on host metabolism is also exemplified by the relation of high-fat diets and choline deficiency. High-fat diet leads to the formation of intestinal microbiota that convert dietary choline into methylamines [128]. This microbiota-related reduction of choline bioavailability may provoke the inability to synthesize phosphatidylcholine, which is necessary for the assembly and secretion of very-low-density lipoprotein (VLDL) [129], inducing the accumulation of triglycerides in the liver. The high-fat diet thus mimics the effect of choline-deficient diets, causing NAFLD [128].

### 7.4. Intestinal Permeability, Endotoxemia and Bacterial Translocation

Miele et al. [65] presented the first evidence of increased intestinal permeability and decreased tight junction protein zonula occludens-1 (ZO-1) expression in biopsy-proven NAFLD patients compared with healthy subjects. Intestinal permeability is increased in children with NAFLD, and correlates with a degree of hepatic involvement [130], as well as blood ethanol and endotoxin levels [131].

Children with NAFLD had significantly higher serum concentrations of endotoxins than controls [132,133]. Patients with NASH revealed higher serum endotoxin levels together with enhanced TLR4 protein expression in the liver compared with patients without NASH [134].

Hepatic TLR4 expression was further proved to be up-regulated in a large cohort of NASH patients when compared with NAFL patients, and this seems to occur in a setting of increased plasma endotoxin and fatty acids [135]. An importance of TLR4 signaling was further suggested by the study reporting that TLR4 codon 299 heterozygous gene mutation (Asp299Gly) was significantly lower in the NAFLD than in the control subjects [136].

Experimentally, genetically obese *ob/ob* and *db/db* mice exhibited enhanced intestinal permeability, profoundly modified distribution of occludin and ZO-1 in the intestinal mucosa together with higher circulating levels of inflammatory cytokines, and portal endotoxemia compared with lean mice [137]. Lastly, growing evidence suggests that at the early course of feeding high-fat diet, not only bacterial products but also complete living bacteria can be translocated from the intestinal lumen towards tissues, including the adipose tissue [138], enhancing the role of BT and gut permeability in tissue injury [66].

The alcohol hypothesis of NASH could explain the similarity of liver histology observed in patients with alcoholic steatohepatitis and NASH and may partly explain the observation of increased gut permeability [65] and endotoxemia [132,133] in NASH patients, because alcohol is known to increase gut permeability [71]. The gene expression of all three major pathways for ethanol catabolism in NASH liver is proved to be highly elevated [139]. Although Volynets et al. [71] observed increased blood ethanol as well as endotoxin levels also in NAFL patients, Zhu et al. [84] found that the blood ethanol concentration was elevated only when obese individuals had NASH.

### 7.5. NASH as a Derangement of Gut-Liver Axis

Mice deficient in PAMPs or downstream signaling are resistant to NASH [140,141]. Dietary habits, by increasing the intestinal gram-negative endotoxin producers, may accelerate liver fibrosis, introducing dysbiosis as a cofactor enhancing chronic liver injury in NAFLD patients [142]. The pathogenetic mechanism of NASH has been recently elucidated from the pathway of gut-liver axis. Rivera et al. [141] first reported the importance of TLR4 signaling in the NAFLD liver. They observed typical steatohepatitis, portal endotoxemia, and increased TLR4 expression in wild-type mice fed MCDD. In contrast, injury and lipid accumulation were significantly lower in TLR4 mutant mice. The destruction of KCs with clodronate liposomes blunted histological evidence of steatohepatitis and inhibited increases in TLR4 expression.

We have reported the enhanced expressions of TNF-α, TLR4, and CD14 mRNA together with reduced phagocytic function of the KCs in the rat fed a choline-deficient l-amino-acid- defined (CDAA) diet [143,144]. We further reported enhanced TNF-α production from isolated KCs and greater expression of TNF-α, TLR4, and macrophage/dendritic cells in the submucosa of ileum in this NASH model [145]. In a clinical situation, microbial products and lipid particles related to intestinal dysbiosis and high-fat diet are considered to reach the liver via the portal vein in patients with NASH. Phagocytic function of KCs is overwhelmed by taking these substances. However, the activation of TLR4-Myd 88 signaling may further proceed without intracellular uptake of endotoxin because TLR4 is the signaling but not the LPS uptake receptor [146]. These may explain the observed dissociation between the depressed phagocytosis and the accelerated TLR4 activation in KCs in our above-mentioned results. Overwhelming endotoxin presumably stimulates TLR4 on the HSC, which enhances the production of extracellular matrix inducing liver fibrosis. Our group further reported exaggerated α-SMA expression (suggesting HSC activation), enhanced liver LBP mRNA levels (suggesting portal endotoxemia), increased intestinal permeability, and depressed intestinal tight junction protein expression in the above rat NASH model [147]. Inhibition of LPS-TLR4 signaling with oral administration of poorly absorbable antibiotics improved all of these intestinal and liver events and suppressed the progression of liver fibrosis [147].

The importance of the gut-liver axis and TLR4 signaling in the progression of NAFLD has been confirmed by other animal studies as well. C3H/HeJ mice, which show a loss-of- function mutation in TLR4, are protected against the development of diet-induced obesity [148]. These mice demonstrate decreased adiposity, increased oxygen consumption, a depressed respiratory exchange ratio, improved insulin sensitivity, and enhanced the insulin-signaling in adipose tissue, muscle, and liver compared with control mice fed high-fat diet [148]. TLR4 and its coreceptor, myeloid differentiation factor-2 (MD-2), are key factors in the recognition of LPS and the activation of proinflammatory pathways. In MD-2 KO and TLR4 KO mice fed MCDD, liver triglyceride accumulation and increased thiobarbiturate reactive substances, a marker of lipid peroxidation, were significantly attenuated [149].

In addition to TLR4, TLR2, and TLR9 are now recognized to play a role in NAFLD pathogenesis [150,151]. TLR9 recognizes DNA containing an unmethylated-CpG motif, which is rich in bacterial DNA [152] detectable in the blood and liver. TLR9 expression is increased in experimental NASH models. [140,150], while TLR9-deficient mice on the CDAA diet showed less steatosis, inflammation, and liver fibrosis compared with the wild type *(WT)* counterparts [140]. TLR2 reacts components of gram-positive bacterial cell walls such as peptidoglycan and lipoteichoic acid [152]. TLR2-deficient mice are known to resist to CDAA-induced steatohepatitis, showing lower expression of proinflammatory cytokines [153].

## 8. Liver Cirrhosis

Liver cirrhosis is a frequent consequence of the long-lasting clinical course of all chronic liver diseases characterized by tissue fibrosis and the conversion of normal liver architecture into cirrhotic nodules [154]. Portal hypertension underlies most of the clinical complications of the disease [154]. Bacterial infections account for significant morbidity and mortality [155] and infections increase mortality four-fold in cirrhotic patients [156]. Although urinary, respiratory, ascitic fluid infections and bacteremia are common infectious complications, spontaneous bacterial peritonitis (SBP) occurs most frequently in these patients. A vast majority of such infections are attributable to enteric gram-negative bacteria, mainly *Enterobacteriaceae* [157]. The investigation of the gut microbiome in cirrhotic patients is important because of the key roles of BT and endotoxemia in the pathogenesis of the above various complications [158].

### 8.1. Endotoxemia, Intestinal Permeability and Bacterial Translocation

Early study using Limulus amebocyte lysate (LAL) test revealed increased incidence of systemic endotoxemia in patients with liver cirrhosis compared with those without liver diseases (48.4%; 15/31 cases vs. 9.5%; 2/21 cases) [159]. The LAL test further detected portal venous endotoxemia in 42.9% (9/21) patients without liver diseases [159]. Subsequent quantitative endotoxin assays disclosed elevated systemic endotoxin levels with the progression of liver cirrhosis [160,161]. Lumsden et al. [162] and Tachiyama et al. [163] found elevated endotoxin levels in the portal blood from cirrhotic patients, which suggested an increased intestinal production and/or absorption of endotoxin in cirrhotic patients, which later opened the discussion on bacterial overgrowth and leaky gut in cirrhosis [164]. Accumulating results have suggested close relations of endotoxemia to cirrhotic complications such as hyperdynamic circulation, portal hypertension, renal, cardiac, pulmonary and coagulation disturbances [1].

Structural and functional changes in the intestinal wall that increase the gut permeability of bacteria and its products have been detected in patients with liver cirrhosis [165]. This gut barrier dysfunction has been considered as an important pathologic factor for several complications of liver cirrhosis [56]. Alcohol drinking, portal hypertension, alterations in the intestinal microbiota, inflammation, and oxidative stress, and endotoxemia can all affect barrier function of both the small and large intestine and may contribute to the development of cirrhotic complications [17,166]. Several authors [53,167,168,169] have reported that patients with liver cirrhosis revealed intestinal hyperpermeability, the state of so-called “leaky gut”. This intestinal hyperpermeability has been reported to be common in cirrhotic patients with a history of SBP [169], associated with the grade of liver cirrhosis as assessed by the Child-Pugh classification [167,168,169], and also has been regarded as a predictor of bacterial infections [170].

Pijls et al. [171] showed that small intestinal permeability determined by urinary lactulose/rhamnose excretion ratio is not altered, whereas large intestinal permeability is increased in patients with stable compensated cirrhosis. As larger numbers of diverse bacteria are present in the large intestine, an increased permeability of this site may enhance the risk of BT [171].

The passage of viable bacteria from the intestinal lumen through the intestinal wall and their translocation to mesenteric lymph nodes and other sites is the accepted pathologic mechanism for the development of spontaneous infections, including SBP or bacteremia [165]. Bacterial products, including endotoxin or bacterial DNA can translocate to extra-intestinal sites and promote an immunological response similar to that provoked by viable bacteria in patients with advanced cirrhosis and ascites [172]. Cirrhotic patients with translocated bacterial DNA of gram-positive taxa show increased pro-inflammatory cytokine levels unrelated to endotoxemia [173]. Pathological BT is thus a contributing factor in the development of complications in cirrhosis, not only in infections, but by exerting a profound inflammatory reaction and exacerbating the hemodynamic derangement [17,165].

### 8.2. SIBO and Gut Dysbiosis

SIBO was diagnosed in 59% of cirrhotic patients by the culture of proximal jejunal aspirations, which was associated with systemic endotoxemia [174]. Although the glucose breath hydrogen test is an alternative method, it shows a limited sensitivity with a positive test ranging from 30 to 38% in cirrhotic patients [56,175,176]. Anyway, the breath test frequently detects SIBO in cirrhotic patients with ascites and advanced liver dysfunction and in those with a history of SBP [175,177]. Delayed gut transit time may be associated with the development of SIBO [56]. Orocecal transit time (OCTT) and small-bowel residence time in cirrhotic patients with SIBO were significantly longer than in those without it [178,179]. Acceleration of orocecal transit by cisapride is reported to abolish bacterial overgrowth in cirrhotic patients [180]. Although the precise mechanism of delayed intestinal transit remains unclear, multiple factors such as the autonomic neuropathy, metabolic derangements, and hyperglycemic state could be implicated [178]. In addition, SIBO itself may lead to delayed intestinal transit [178], for antibiotic therapy has been found to reduce the OCTT in cirrhotic patients [181]. SIBO in cirrhosis showed a high correlation with the presence of bacterial DNA fragments in peripheral blood, suggesting that SIBO could be a major risk factor of BT, especially in ascitic patients [182].

A culture-based study in China [183] first reported enriched pathogenic *Escherichia coli* and *Staphylococcal spp.* in the feces of patients with liver cirrhosis (mostly associated with hepatitis B virus infection) and minimal hepatic encephalopathy (HE). In 2011, non-culture 16S ribosomal RNA-based pyrosequencing analysis of feces found reduced microbial diversity and marked dysbiosis in Chinese patients with cirrhosis associated with hepatitis B virus (HBV) infection and alcohol drinking. [38]. Although the involved microbial taxa are not consistent among different population groups, a shortage of autochthonous non- pathogenic bacteria and an overgrowth of potentially pathogenic bacteria are common findings [166]. This study [38] further reported that the proportion of phylum *Bacteroidetes* was decreased, although *Proteobacteria* and *Fusobacteria* were increased. The authors later found decreases in the genera *Eubacterium*, *Bacteroides* and *Alistipes* and increases in the genera *Veillonella*, *Clostridium*, *Streptococcus*, and *Prevotella* [98]. Among the markedly enriched 20 species, 6 were *Veillonella spp*. and 4 were *Streptococcus spp.* [98]. Yet another study from this Chinese study group [184] showed that patients with HBV-related cirrhosis exhibited decreases in *F. prausnitzii*, *Bifidobacterium*, and *Clostridium* and increases in *Enterobacteriaceae* and *Enterococcus faecalis*. *F. prausnitzii* is an anti-inflammatory bacterium that stimulates interleukin (IL)-10 secretion and inhibits IL-12 and interferon- gamma expressions [35]. They further observed a reduction of *Bifidobacterium*, especially *Bifidobacterium catenulatum*, in the feces of HBV-related cirrhotics, [185] and a marked decrease in *Lactobacillus rhamnosus* and a reduction in *Lactobacillus fermentus* in the feces of decompensated cirrhotics [186]. Another Chinese study group also reported that the feces of HBV-related liver cirrhosis showed increased *Proteobacteria* and decreased *Bacteroidetes*, presenting enriched *Escherichia coli*, *Veillonella dispar*, and *Veillonella parvula* and reduced *Bacteroides* [187].

On the other hand, Bajaj et al. [188] reported that fecal microbiota in American patients with alcohol-related and hepatitis C virus (HCV)-related liver cirrhosis had a lower proportion of *Ruminococcaceae*, *Lachnospiraceae* and *Clostridiales Incertae Sedis XIV* and a higher proportion of *Leuconostocaceae*, *Enterobacteriaceae*, and *Fusobacteriaceae* compared with healthy subjects. *Ruminococcaceae*, *Lachnospiraceae* and *F. prausnitzii* are butyrate-producing bacteria. As described previously, butyrate is an important energy source for enterocytes, protecting the intestinal barrier through the stimulation of tight junctions and mucous production [9,189,190]. SCFAs are also known to maintain normal colonocyte turnover, to increase anti-bacterial peptides (LL-37 and CAP-18) and to mitigate colonic inflammation [190,191]. They have various anti-inflammatory effects both on the innate and adaptive immune systems, suppressing inflammatory cytokine production, inhibiting immune cell maturation and recruitment, and inducting regulatory T cells [191,192]. A reduction in the SCFA-producing *Lachnospiraceae* might result in an elevation of colonic pH, which is considered to enhance the production and absorption of ammonia and provoke HE [38,60].

In contrast to the previous reports from China, Dubinkina et al. [37] found that fecal microbiome expressed enrichment in several *Bifidobacterium* (*B. longum, dentrium, and breve*) and *Lactobacillus* spp. (*L. salivarius, antri, and crispatus*) in Russian patients with alcoholic dependence and liver cirrhosis. They discussed that the expansion of these microbiome suggested that uniform probiotic interventions using these genera should be cautious for cirrhotic patients.

When cirrhotic patients were divided into 3 groups based on the severity of liver cirrhosis (compensated outpatients, decompensated outpatients and inpatients), there was a progressive reduction in autochthonous taxa, *Clostridiales XIV*, *Ruminococcaceae* and *Lachnospiraceae,* and a progressive increase in pathogenic taxa, *Enterococcaeae* and *Enterobacteriaceae* with worsening liver cirrhosis [40]. *Veillonellaceae* and *Porphyromonadaceae* were also reduced with worsening of liver cirrhosis [40]. As an index presenting the grade of gut dysbiosis, they proposed cirrhosis dysbiosis ratio (CDR), which is the ratio of the amounts of beneficial autochthonous taxa (*Lachnospiraceae* + *Ruminococaceae* +*Veillonellaceae* + *Clostridiales Incertae Sedis XIV*) to those of potentially pathogenic taxa (*Enterobacteriaceae* + *Bacteroidaceae*) [40]. This CDR was inversely correlated to the model of end-stage liver disease (MELD) score and the blood endotoxin level [40]. A low CDR was also associated with death and organ failure within 30 days [40]. In this study, patients with acute-on-chronic liver failure (ACLF) showed higher serum endotoxin and lower CDR due to decreased *Clostridiales XIV* and *Leuconostocaceae* than those without ACLF [40]. Taken together, changes in fecal microbiota have been linked with 90-day hospitalizations, organ failure, and death in hospitalized patients with cirrhosis [193].

On the other hand, Liu et al. [194] reported that the grade of gut dysbiosis may depend on small bowel transit (SBT) rather than the severity of liver cirrhosis. They confirmed gut dysbiosis evaluated by the *Firmicutes/Bacteridetes* ratio and the microbial dysbiosis index (MDI) [195] (the log of total abundance of organisms increased divided by total abundance of organisms decreased) were negatively correlated to small bowel transit (SBT) rather than Child–Pugh score [194].

Recent study has additionally shed light on the role of gut fungi, that may affect the pathophysiology of liver cirrhosis and impact hospitalizations in conjunction with bacterial dysbiosis [193]. Fungal dysbiosis is characterized by a lower fungal diversity and a decrease in *Basidiomycota*/*Ascomycota* ratio together with an increase in *Ascomycota* including *Candida* is observed in the feces of patients with advanced and infected cirrhosis [196]. It could be exacerbated by prevailing antibiotic use and may provoke fungal infection and impact hospitalization in close relation to bacterial dysbiosis in cirrhotic patients [196]. 

Dysbiosis with reduced autochthonous taxa and increased potentially pathogenic families (*Enterobacteriaceae*, *Enterococcaceae*) is found in saliva of cirrhotic patients, especially those with previous history of HE or hospitalizations [197]. The salivary microbiome also has been correlated with liver-related 90-day hospitalization independent of MELD score and HE state [198]. Qin et al. [98] reported that most bacteria found in the feces of cirrhotic patients was of buccal origin, suggesting invasion of the gut by oral bacterial species. Interestingly, the above-mentioned, presumably pathogenic *Streptococcus spp.* and *Veillonella spp.* are recognized to be of oral origin [98]. The invading bacteria are found not only in the colon but also in the ileum and duodenum, which may potentiate SIBO in patients with liver cirrhosis [98]. The mechanism of this invasion has not been fully clarified but may be related to impaired bile acid (BA) and gastric acid output in cirrhosis [199]. It is well known that gastric acid and bile acids act to control bacterial colonization, attachment, and infiltration into the host [200]. In patients with decompensated cirrhosis, an administration and a withdrawal of proton pump inhibitor omeprazole were reported to result in an increase and a decrease in oral-origin microbial taxa (i.e., the relative abundance of *Streptococcaceae*) in the feces, respectively [201]. As for oral infection, the patients with periapical radiolucency, a sign of apical periodontitis and radicular cysts, had a higher prevalence of cirrhosis-related complications such as ascites, HE, and variceal bleeding [202,203].

### 8.3. Fecal and Mucosal Dysbiosis in Cirrhotic Complications

Several studies have highlighted the changes in gut microbiome in patients with liver cirrhosis and HE. Liu et al. [183] studied patients with liver cirrhosis (mostly attributed to HBV and HCV infection) with minimal HE and reported an evident fecal overgrowth of potentially pathogenic *Escherichia coli* (*E. coli*) and *Staphylococcus* spp. in the feces. A non-culture analysis revealed that urease containing bacteria *Streptococcus salivarius*, absent in the feces of normal subjects, significantly increased in the feces of cirrhotic patients with mild HE [204]. The amount of this bacteria was positively correlated with circulating ammonia levels in cirrhotic patients [204]. Bajaj et al. [188] on the other hand showed that American patients with cirrhosis and HE had a higher fecal abundance of *Enterobacteriaceae, Streptococcaceae, Fusobacteriaceae, Leuconostocacea,* and *Alcaligenaceae* compared with control subjects. They additionally found *Veillonellaceae* was the only significantly increased taxa in the feces of cirrhotics with HE compared with those without HE [188] In their cirrhotic patients, *Alcaligeneceae* and *Porphyromonadaceae* were positively correlated with cognitive impairment [188]. With regard to the pathogenesis of HE, accumulating evidence indicates that hyperammonemia, systemic inflammation and neuroinflammation related to endotoxemia and gut dysbiosis play cardinal roles.

In line with these results, Bajaj et al. [188] showed that *Fusobacteriaceae*, *Veillonellaceae* and *Enterobacteriaceae* were positively, and *Ruminococcaceae* were negatively correlated to inflammatory cytokines in the blood [188]. Network analysis revealed strong correlations between the microbiome, cognition and inflammatory cytokines, only in cirrhotic patients with HE, [188]. Ahluwalia et al. [205] reported that fecal *Enterobacteriaceae* was positively correlated and autochthonous taxa was negatively correlated with hyperammonemia- associated astrocytic changes on magnetic resonance spectroscopy and that *Porphyromonadaceae* was correlated with neuronal changes on diffusion tensor imaging. The relation of *Enterobacteriaceae* and *Alcaligeneceae* to HE may be partly explained by the fact that these bacteria degrade urea to produce ammonia [188,206,207]. In addition, *Enterobacteriaceae* is a known organism that produces potent endotoxin. Sung et al. in Taiwan [208] reported that the relative abundance of Bacteroidetes phylum in the microbiome decreased, whereas that of *Firmicutes*, *Proteobacteria* and *Actinobacteria* increased in cirrhotic patients during acute hepatic encephalopathy (AHE) compared with those with compensated cirrhosis. Of them, the abundance of *Veillonella parvula* increased the most during AHE via a metagenomics recovery of the genomes. Its abundance in AHE patients is 10 times that of control cirrhosis patients [208]. Moreover, the relative abundances of three (*Alistipes*, *Bacteroides*, *Phascolarctobacterium*) and five OTUs (*Clostridium-XI*, *Bacteroides*, *Bacteroides*, *Lactobacillus*, *Clostridium-sedis*) at AHE were associated with HE recurrence and overall survival, respectively, during the one-year follow-up [208].

The significance of *Veillonellaceae* in decompensated cirrhosis is still undetermined at present, because Bajaj et al. [40] later reported its reduction with the progression of liver cirrhosis and used it for calculation of CDR as beneficial autochthonous bacteria [40].

Bacterial infections (i.e., respiratory, urinary and ascitic fluid infections) are important risk factor for early mortality in cirrhotic patients and are mostly related to enteric Gram- negative bacteria, mainly *Enterobacteriaceae* [156,157]. Bacterial DNA was reported to be detected in blood and ascites in one-third of cirrhotic patients with portal hypertension and culture-negative ascites [209]. Most bacteria causing SBP are Gram-negative bacilli, especially *Enterobacteriaceae* family (*E. coli*, *Enterobacter, Klebsiella and Proteus*), which are predominant in the gut microbiota of cirrhotic patients [38,184,188,210,211]. Bajaj et al. [40] found that fecal microbiota in cirrhotic patients admitted with infections showed significant differences in the fecal microbiota characterized by increased *Enterobacteriaceae* and decreased *Coriobacteriaceae*, *Lachnospiraceae*, *Ruminococcaceae*, *Veillonellaceae* and *Clostridiales XIV*, lower CDR and higher plasma endotoxin levels compared with those without infections. This study additionally described that the routine culture detected *Streptococcus spp.*, *Klebsiella*, *Escherichia* and *Citrobacter spp.* for SBP and *E. coli*, *Enterococcus*, *Staphylococcus aureus* and *Lactococcus* for urinary tract infections [40].

Simultaneous presence of bacterial DNA in blood and ascites was associated with SBP, blood stream infections and markers of inflammation [212]. Further, bacterial DNA of Gram- negative species was found in ascitic fluid with the increasing severity of multi-organ failure in ACLF [212]. Taken together, emerging evidence revealed that the gut dysbiosis and bacterial translocation drive the progression of severe infection and hepatic failure in liver cirrhosis.

The microbiota inside the mucus layer adherent to the mucosal epithelium is more stable than the luminal counterpart and constantly cross-talks with the host [213]. Bajaj et al. [214] examined the recto-sigmoidal mucosal microbiome and the fecal microbiome, as the former may reflect the intestinal barrier condition more precisely. They found that the recto-sigmoidal mucosal microbiome in cirrhotic patients had a lower abundance of autochthonous bacteria (*Subdoligranulum*, *Dorea*, and *Incertae Sedis XIV other*) and a higher abundance of potentially pathogenic bacteria (*Enterococcus*, *Clostridium*, *Burkholderia*, and *Proteus*) compared with healthy controls. They further noted that cirrhotics with HE had lower *Roseburia* and higher *Enterococcus*, *Veillonella*, *Megasphaera,* and *Burkholderia* in the mucosa compared with those without HE, whereas fecal microbiome composition was not different between the two groups in this study [214]. This mucosal dysbiosis is also recognized to imply a disease-associated reduction in the autochthonous bacteria and the increased pathogenic genera in cirrhotic patients with HE [214]. On a correlation network analysis, these overrepresented genera were linked to poor cognitive performance, disease severity (MELD scores), enhanced systemic inflammation (serum IL-17 levels), and endothelial activation (serum soluble intravascular adhesion molecule (sICAM-1) levels) [214]. Concomitant evaluations of microbiome in the cecal mucosa, feces, and peripheral/portal blood from patients with liver cirrhosis by another study group revealed that the portal blood had a bacterial community composition similar to that of the colonic mucosa, but not to that of the feces, strengthening the importance of mucosa-associated microbiota for studying pathophysiological meanings in liver cirrhosis [215]. Chen et al. [216] examined the duodenal mucosal microbe and reported that *Veillonella*, *Megasphaera*, *Dialister*, *Atopobium* and *Prevotella* were increased in cirrhotic patients compared with healthy subjects whose duodenal mucosa enriched with *Neisseria*, *Haemophilus* and *SR1 genera incertae sedis*. This deviation of duodenal microbiota may impact on the inflammatory changes, because SIBO in possible association with duodenal dysbiosis has the greatest potential for promoting BT in patients with liver cirrhosis [216]. The enriched microbiota in duodenal mucosa, commonly found in the oral cavity, are consistent with the oral microbiota, which produce high levels of methanetiol (CH_3_SH), an important factor in the pathogenesis of HE [216].

### 8.4. Metabolic Changes Related to Gut Microbiota

Cirrhotic patients, compared with controls, had a higher *Enterobacteriaceae* (potentially pathogenic) but lower *Lachonospiraceae*, *Ruminococcaceae*, and *Blautia* (7α- dehydroxylating bacteria) abundance. The gut 7α-dehydroxylating bacteria are known to convert primary BAs (CDCA and CA), into secondary BAs (LCA and DCA, respectively). Kakiyama et al. [217] analyzed fecal microbiota together with BAs and found that CDCA was positively correlated with *Enterobacteriaceae* and DCA was positively correlated with *Ruminococcaceae*. They further showed a positive correlation between *Ruminococcaceae* and DCA/CA and a positive correlation between *Blautia* and LCA/CDCA. These relations suggest that a decreased conversion of primary to secondary BAs is linked with an abundance of key gut microbiome in advanced liver cirrhosis. BT induces inflammation, which suppresses synthesis of total BAs in the liver via the inhibition of CYP7A1. BAs prevent BT and attenuate the passage of bacterial products from the lumen of the intestine [165,218,219]. A decrease in BAs entering the intestines favors overgrowth of pathogenic and proinflammatory microbiome including *Porphyromonadaceae* and *Enterobacteriaceae* [12]. Kakiyama et al. [39] further reported that actively drinking patients with cirrhotic exhibited increased expression of FXRα mRNA in the ileum and sigmoid colonic mucosa together with increased expression of TNF-α, IL-1β, IL-6 and MCP-1 mRNA in the colonic mucosa.

New strategies that specifically targets BA receptors may open new doors in the treatment of liver cirrhosis. Renga et al. [220] showed that in vivo activation of FXR regulates the expression of genes involved in glutamine/glutamate metabolism and stimulates urea synthesis and ammonia detoxification in a rodent model of cirrhosis. Verbeke et al. [221] showed that FXR agonist obeticholic acid improves portal hypertension in two different rat models of cirrhosis by decreasing intrahepatic vascular resistance by increasing intrahepatic eNOS activity. They further found that this FXR agonist prevents gut barrier dysfunction, intestinal inflammation, and BT in cholestatic rats, and thus demonstrated a crucial protective role of FXR in the gut-liver axis [222].

The metagenomic approach to intestinal microbiota may be of particular importance for understanding the pathophysiology of liver cirrhosis, where variable metabolic and immunologic changes play cardinal roles. Metagenomic pyrosequencing of intestinal microbiota enables researchers to discover novel genes from uncultivated microorganisms, and to assemble whole genomes from community DNA sequence data [223]. A functional gene array in patients with alcoholic and HBV-related cirrhosis revealed that the functional composition of fecal microbiomes was mostly affected by alcohol consumption, and secondly by cirrhosis [223]. Alcohol consumption resulted in enrichment of functional genes including xenobiotic metabolism and virulence, while both cirrhosis groups were markedly depleted in the functional genes involved in nutrient processing related to amino acids, lipids and nucleotides [223]. To explore further the microbial genes associated with liver cirrhosis, Qin et al. [98] grouped the microbial genes into clusters and denoted metagenomic species based on their abundance profiles. At the module or pathway level, the liver cirrhosis-associated markers included assimilation or dissimilation of nitrate to or from ammonia, denitrification, γ-aminobutyric acid (GABA) biosynthesis, GABA shunt, hem biosynthesis, phosphotransferase systems, and membrane transport of amino acids. In contrast, the prevalent markers in control controls were those involved in the metabolism of carbohydrates, amino acids, cofactors and vitamins as well as energy metabolism and signal transduction. The enriched modules for ammonia and GABA production in cirrhotic patients suggest a potential role of gut microbiota in HE [98]. Iebba et al. [215] reported that fecal *Enterobacteriaceae* and trimethylamine were positively correlated with blood proinflammatory cytokines, while *Ruminococcaceae* and SCFAs played a protective role. Within the peripheral blood and feces, *Stenotrophomonas pavanii*, *Methylobacterium extorquens* as well as methanol and threonine were positively related with HE.

In contrast to these studies, Wei et al. [224] surprisingly noticed increased bacterial metabolic activities of carbohydrate, branched-chain amino acid, pantothenate and CoA with the worsening of cirrhosis from Child-Pugh class A to class C by their metaproteome analysis. They thought that the changes in intestinal microenvironment in patients with liver cirrhosis enhanced the growth and the protein expression of gut microbiome. The findings suggested that the fecal microbiome could have adaptability to the intestinal microenvironment and worked to compensate for the innutritious body of cirrhotic patients [224]. In line with their concept, the kind of bacteria responsible for the supposed metabolic compensation is of utmost importance. Their results are contradictory to the prevailing concept that gut dysbiosis cause metabolic disturbances in liver cirrhosis and arouse a question of whether we can really improve the metabolic state of advanced cirrhosis by correcting gut dysbiosis [116]. This hypothesis is interesting but needs to be further explored. Changes in intestinal microbiota associated with clinical studies on liver cirrhosis. Intestinal microbiota in patients with liver cirrhosis is summarized in Table 3.

### 8.5. Gut Microbiome before and after Liver Transplantation

An early study [225] reported that the liver transplantation (LT) recipients with HBV-related cirrhosis had fecal dysbiosis characterized by the decreases in *Bifidobacterium spp.*, *Lactobacillus spp.*, and *Faecalibacterum prausnitzii* and the increases in *Enterobacteriaceae* and *Enterococcus spp.* after LT and that the microbiota was tended to restore normal pattern except for *Enterococcus spp.* over time [225]. Bajaj et al. [226] noted a significant improvement in cognition with increase in fecal microbial diversity, increase in autochthonous and decrease in potentially pathogenic genera belonging to *Enterobacteriaceae* (*Escherichia*, *Shigella*, *Salmonella*), and reduced endotoxemia were seen in cirrhotic patients with good clinical course at 7 ± 3 months after LT compared with baseline. In the meanwhile, potentially beneficial, autochthonous genera belonging to *Ruminococcaceae* and *Lachnospiraceae* were increased. Sun et al. [227] reported reduced fecal abundance of *Actinobacillus*, *Escherichia*, and *Shigella* and increased abundance of *Micromonosporaceae*, *Desulfobacterales*, *the Sarcina genus of Eubacteriaceae*, and *Akkermansia* after LT. Improvement of gut microbiota was accompanied by favorable changes in gut microbial functionality corresponding to BAs (higher secondary, oxo and iso-BAs), ammonia, endotoxemia, lipidomic, and metabolomic profiles in LT recipients [228].

Although the changes in microbiota after LT were different in detail, the trend of reducing pathological taxa and recovering autochthonous taxa is common among the studies. Bajaj et al. [226] found higher *Bacteroidaceae abundance* and lower beneficial *Clostridial families* (*Ruminococcaceae*, *Lachnospiraceae*) and *Bifidobacteriaceae* in post-LT samples, despite a statistically similar microbial diversity to control samples. Posttransplant cognitive impairment was proved to be related to residual dysbiosis characterized by higher relative abundance of *Proteobacteria* (family *Enterobacteriaceae*) [226].

Concerning with the relation of gut microbiome before LT and the post-LT complication, cirrhotic patients with decreased microbial diversity before LT were more likely to develop postoperative infections than those with stable microbiota consisting of slight microbial dysbiosis [229,230]. Other clinical and experimental studies have also demonstrated that the gut microbiota disturbance, especially loss of diversity, can be associated with infection and rejection risk after LT. [226,231,232].

However, modulation of gut microbiota by antibiotics with pre-/probiotics to prevent post- LT complication management requires further investigation [230].

## 9. Primary Sclerosing Cholangitis (PSC)

Primary sclerosing cholangitis (PSC) is a cholestatic liver disease characterized by stricture of the intra- and/or extrahepatic bile ducts [233]. Approximately 70% of patients with PSC also have inflammatory bowel disease (IBD), mainly ulcerative colitis, whereas only 2% to 8.1% of patients with IBD have PSC [233]. PSC-associated IBD is associated with a higher risk of colorectal cancer [234].

Autoantibodies against biliary epithelial cells (BECs) can be found in 63% of patients with PSC [235]. These anti-BEC antibodies can induce the expression of CD44 and the production of proinflammatory IL-6 in BEC, which may enhance the destruction and inflammation of BEC [235]. These autoantibodies include atypical perinuclear antineutrophil cytoplasmic antibodies (p-ANCAs) that are directed against human autoantigen beta- tubulin isotype 5 (TBB-5) and its bacterial homologue FtsZ, which is expressed by almost all commensal bacteria [236,237]. The binding of autoantibodies to BEC results in an upregulation of TLR expression, which in turn might sensitize the biliary tract to microbial products [237,238].

Patients with PSC had markedly reduced bacterial diversity compared with healthy control independent of the presence of IBD [239,240,241,242], and a different global microbial composition compared with both healthy subjects and patients with ulcerative colitis (UC) [240]. Rossen et al. [239] noted that the relative abundance of uncultured *Clostridiales II* in the ileocecal mucosa was significantly lower in 12 PSC patients compared with UC patients and control subjects by 16S rRNA based analyses. Marked depletion of this bacterial group, considered to be identical with *Christensenella*/*Catabacter* group, was reported in fecal samples of patients with post-infectious irritable bowel syndrome [243,244]. Torres et al. [245] on the other hand found that 20 American PSC patients complicated with IBD was characterized by enrichment of *Blautia* and *Barnesiellaceae* in the microbiome of colonic mucosa compared with IBD patients.

Sabino et al. [241] reported an overrepresentation of *Enterococcus*, *Fusobacterium* and *Lactobacillus* genera in the feces of 66 Belgian patients with PSC irrespective of concomitant IBD. Their study indicated that *Enterococcus* and *Lactobacillus* were increased in PSC patients without liver cirrhosis, LT or concomitant IBD [241], although the dysbiosis had been associated with liver cirrhosis [38,98] and IBD [195]. The increase in *Fusobacterium*, also reported later by Torres et al. [246], might be meaningful, when considering its known linking with colorectal cancer in PSC patients [247]. nishiishi et al. [248] demonstrated significant increases in *Escherichia*, *Lachnospiraceae* and *Megasphera* in the patients with PSC–IBD compared with controls. Bajer *et al.* [249] found fecal *Rothia*, *Enterococcus*, *Streptococcus*, *Clostridium*, *Veillonella* and *Haemophilus* were increased in 43 Czech patients with PSC regardless of concomitant IBD compared with healthy subjects. Kummen et al. [240] stressed that *Veillonella* genus in the feces showed a marked increase in 85 Norwegean PSC patients compared with both healthy controls and UC patients and thought it could potentially influence disease progression [250]. In a letter to the editor, Rühlemann et al. [251] also admitted the enriched *Veillonella* in their 73 northern German patients with PSC compared with healthy controls. But in contrast to the study by Kummen et al. [250], they could not find the difference in *Veillonella* between PSC patients and UC patients and insisted that the abundances of four taxa (*Veillonella*, *Clostridiales*, *Lachnospiraceae* and *Coprococcus*) was more useful for discriminating PSC from healthy subjects than the abundance of the genus *Veillonella* only [251]. Another letter to the editor by Iwasawa et al. [252] reported increased *Enterococcus*, and decreased *Parabacteroides* in the feces of 27 Japanese children with PSC and IBD compared with age-matched healthy controls. The abundance of *Faecalibacterium*, *Ruminococcus* and *Roseburia* were higher in these patients than age-matched children with UC [252]. The elevation of these three genera, which include many species producing an anti- inflammatory butyrate, may be related to biliary inflammation in PSC children.

Recently Torres et al. [246] reported increased abundance of fecal *Ruminococcus* and *Fusobacterium* in 15 Portuguese PSC-IBD patients compared with IBD patients, while *Dorea*, *Veillonella*, *Lachnospira*, *Blautia*, and *Roseburia* were less abundant. Although the authors explained their inconsistent results on the abundance of *Blautia* in the mucosa and feces by the dissociation between the mucosal microbiota and fecal microbiota [246].

Butyrate-producing bacteria *Blautia* is also important in converting primary BAs to secondary BAs in the intestine by its 7α-dehydroxylation activity. Its impact on the pathogenesis of PSC should be further evaluated. The authors discussed that unexpectedly low abundance of *Veillonella* in their PSC-IBD patients could be explained by the earlier stage of disease [246]. As described in former sections, increased abundance of *Veillonellaceae* and *Veillonella* have been associated with higher systemic inflammation, endotoxemia and HE in patients with liver cirrhotics [188,214,216,246]. It is noteworthy in this study that the amount of *Veillonella* showed strong correlations to multiple bile acid profiles in the feces (e.g., total primary and secondary BAs levels, primary BAs/secondary BAs ratio, and sulfo-conjugates) [246]. *Veillonella* is also known as a 7α-dehydroxylating bacteria, promoting secondary BA formation in the intestine [253]. The serum bile-acid pool was increased, but the stool bile-acid pool was decreased in patients with PSC-IBD compared to patients with IBD alone [231,246]. Further studies should focus on a more comprehensive analysis of *Veillonella* and BA composition in different disease stages. Intestinal dysbiosis in patients with primary sclerosing cholangitis is summarized in Table 4.

## 10. Primary Biliary Cholangitis (PBC)

Primary biliary cholangitis (PBC) is an immune-mediated hepatobiliary disorder characterized by progressive non-suppurative destruction of small bile ducts, resulting in intrahepatic cholestasis, fibrosis, cirrhosis and ultimately end-stage liver failure [254,255]. PBC affects mostly middle-aged women, clinically with symptoms of fatigue and pruritus in the context of chronically elevated alkaline phosphatase (ALP) levels [256]. At present, the standard treatment for PBC is ursodeoxycholic acid (UDCA) [254], which has been reported to delay the progression of liver fibrosis and even relieve fibrosis of cirrhotic patients [257].

Some studies supported its beneficial effect on prognosis [257]. A Japanese study revealed that patients with PBC showed significant increases in *Eubacterium* and *Veillonella* and a significant decrease in *Fusobacterium* in the oral microbiota compared with the healthy controls [258]. In this report, the relative abundance of *Veillonella* was positively correlated with the levels of IL-1β, IL-8 and immunoglobulin A in saliva and the relative abundance of *Lactobacillales* in feces [258].

A first case-control study [259] reported that the fecal microbiome of Chinese patients with early PBC were depleted of potentially beneficial bacteria, such as *Acidobacteria*, *Lachnobacterium sp*., *Bacteroides eggerthii* and *Ruminococcus bromii* compared with healthy controls. The feces of patients were conversely enriched in bacteria containing opportunistic pathogens, such as *gamma-Proteobacteria*, *Enterobacteriaceae*, *Neisseriaceae*, *Spirochaetaceae*, *Veillonella*, *Streptococcus*, *Klebsiella*, *Actinobacillus pleuropneumoniae*, *Anaeroglobus geminatus*, *Enterobacter asburiae*, *Haemophilus parainfluenzae*, *Megasphaera micronuciformis* and *Paraprevotella clara* [259]. On the other hand, another Chinese study on treatment-naive PBC patients compared with healthy controls demonstrated that the patients were associated with altered composition and function of gut microbiota, as well as a moderately lower level of diversity [260]. Their 16S rRNA sequence analyses of fecal microbiota revealed that *Bacteroidetes spp* were decreased, whereas *Fusobacteria* and *Proteobacteria spp* were enriched in PBC [260]. At the genus levels, *Haemophilus*, *Veillonella*, *Clostridium*, *Lactobacillus*, *Streptococcus*, *Pseudomonas*, *Klebsiella* and an unknown genus in the family of *Enterobacteriaceae* (*Enterobacteriaceae,g*) were enriched, while *Sutterella*, *Oscillospira* and *Faecalibacterium* were decreased in PBC patients compared with healthy controls [260]. As serum anti-gp210 antibody has been considered as an index of disease progression, relatively lower species richness and lower abundance of *Faecalibacterium spp*. in gp210-positive patients compared with gp210-negative patients are interesting [260].

## 11. Hepatocellular Carcinoma (HCC)

The insight on the role of gut microbes in hepatocarcinogenesis mainly comes from animal experiments [261,262,263,264], suggesting that the intestinal microbiota and the TLR4-LPS pathway seem to be important for HCC promotion [213,262,264]. Accumulating evidence further support that the promoting mechanisms include dysbiosis-induced altered bacterial metabolites such as DCA, a known promotor of hepatocarcinogenesis, as well as a leaky gut and resulting BT, which enhances chronic hepatic inflammation via TLR-mediated signals just as in the progression of various chronic liver diseases [265].

The clinical studies suggesting the linkage of gut dysbiosis with HCC are scarce and inconclusive [266]. In a culture-based study, Grat et al. [267] compared fecal microbiome between 15 patients with HCC and 15 non-HCC patients who received LT and reported that the presence of HCC was associated with a greater abundance of fecal *Escherichia coli* (*E. coli*). *E. coli* is reported to produce potent endotoxin and to deconjugate conjugated BAs thereby enhancing the formation of secondary BAs. Animal studies indicate that endotoxin and secondary BAs are probable key players to promote HCC in these situations [266]. In patients with NAFLD-associated liver cirrhosis, Ponziani et al. [268] found that the fecal microbiota of patients with HCC was enriched with *Bacteroides*, *Ruminococcaceae*, *Enterococcus*, *Phascolarctobacterium*, and *Oscillospira* and deficient in *Bifidobacterium* and *Blautia* compared with those without HCC. The cirrhotic patients both with and without HCC showed enriched fecal *Enterobacteriaceae* and *Streptococcus* and reduced *Akkermansia* compared with healthy controls [268]. In their study, the increase in fecal *Bacteroides*, which had been reported in NAFLD patients with marked fibrosis [95], was related with increased levels of the proinflammatory cytokines IL8 and IL13, circulating activated monocytes, and monocytic myeloid-derived suppressor cells, suggesting a co-factorial role in hepatocarcinogenesis [268]. They also reported that *Akkermansia* and *Bifidobacterium* were negatively correlated with fecal calprotectin concentration, a surrogate marker of intestinal inflammation in cirrhotic patients [268]. In a trial to evaluate the potential of fecal dysbiosis as non-invasive biomarkers for detecting HCC in patients with HBV-related liver cirrhosis, Ren et al. [269] reported that phylum *Actinobacteria* and 13 genera including *Gemmiger* and *Parabacteroides* were enriched in early HCC versus cirrhosis. Interestingly, gut protective *Akkermansia* and butyrate-producing genera such as *Clostridium IV*, *Ruminococcus*, *Coprococcus*, *Oscillibacter*, *Alistipes*, *Butyricoccus* and *Eubacterium* were decreased, while genera producing-lipopolysaccharide such as *Klebsiella* and *Haemophilus* were increased in their patients with early HCC versus healthy controls [269,270], the phylum Verrucomicrobia and the genera *Alistipes, Phascolarctobacterium* and *Ruminococcus* were decreased substantially while *Klebsiella* and *Haemophilus* were increased in early HCC patients involved in a clinical study [270,271]. They also added that obesity induces changes in the composition of the gut microbiota and its metabolites (LPS and PAMPs) [271].

## 12. Conclusions

Accumulating evidence supports that gut dysbiosis may relate to various liver diseases. Dietary factors seem very important in this new developing, interesting clinical field.

Alcoholics with high intestinal permeability had intestinal dysmotility and altered immune responses. Alcohol-induced dysbiosis may be associated with gut barrier dysfunction. Microbiota and their products modulate intestinal barrier function by affecting epithelial pro-inflammatory responses and mucosal repair functions. Gut dysbiosis frequently causes endotoxemia in alcoholics. This also affects many NAFLD patients. Gut dysbiosis may facilitate energy-producing and proinflammatory condition for the progression of NAFLD. Close relations of endotoxemia is also related to cirrhotic complications where a shortage of autochthonous non-pathogenic bacteria and an overgrowth of potentially pathogenic bacteria are common. The lower ratio of beneficial autochthonous taxa to potentially pathogenic taxa was associated with early death and organ failure for cirrhotics. Cirrhotic patients with decreased microbial diversity have poor prognosis after liver transplantation. Cirrhotic patients with decreased microbial diversity before liver transplantation. Patients with PSC also had marked reduction of bacterial diversity. Treatment-naive PBC patients has marked dysbiosis. The gut dysbiosis-induced deranged gut liver axis may further induce hepatocarcinogenesis. Although the notion that gut dysbiosis may provoke increased IP and the following various liver injuries still a hypothesis based on various clinical and experimental investigations. Although dysbiosis- induced altered bacterial metabolites may still have much problems to be solved, best answer will be found about gut dysbiosis and clinical hepatology after persisting research activities in the near future.

## Figures and Tables

**Figure 1 diseases-07-00058-f001:**
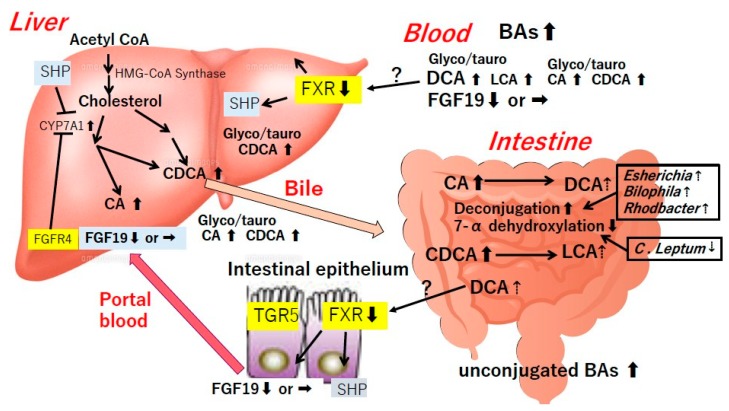
Signaling in the development of nonalcoholic fatty liver disease (NAFLD). The G protein-coupled receptor 43 (GPR43), a receptor for short chain fatty acids (SCFAs), is also termed free fatty acid receptor 2 (Ffar2) in the colonic mucosal cells. It stimulates peptide YY release from neuroendocrine L cells in the intestine. It slows gastric emptying and intestinal transit and enhances nutrient absorption. These L cells also release glucagon-like peptide 1 (GLP-1), which increases glucose-dependent insulin secretion. The secondary to primary BA ratio in the stool was lower in NASH compared with healthy control, but ratio of conjugated to unconjugated BAs was not different between the groups.

**Figure 2 diseases-07-00058-f002:**
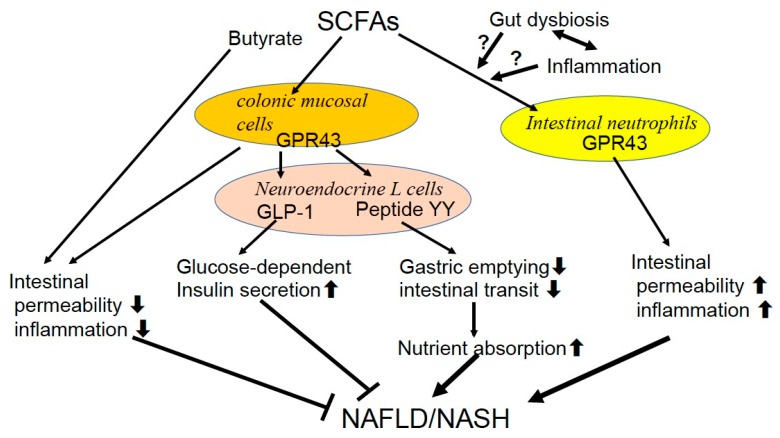
Proposed relationships between gut microbiome.

**Table 1 diseases-07-00058-t001:** Changes in intestinal microbiota associated with clinical studies on alcoholic liver disease (ALD).

Phylum	Class	Order	Family	Genus (Species)
Firmicutes	Bacilli ↑ [33] *	Bacillales	Bacillaceae	Bacillus ↑ [33] *
		Lactobacilales	Lactobacillaceae	Lactobacillus ↓ [32] (L. salivarius ↑ [37])
			Streptococcaceae	Streptococcus ↑ [49] (S. infantarius subsp. Infantarius ↓ [37])
				Lactococcus ↑ [37] (L. lactis subsp Cremoris ↑ [37])
			Enterococcaceae	Enterococcus ↓ [32]
	Clostridia ↓ [33] * [34] ^†^	Clostridiales	Clostridiaceae	Clostridium ↓ [34] ^†^ [33] *,
				Subdoligranulum ↓ [33] *,
			Clostridales incertae sedis XIII ↓ [34] ^†^	
			Clostridales incertae sedis XIV ↑ [34] ^†^ ↓ [40]	
			Eubacteriaceae	Eubacterium (E. eligenes ↓ [37]
			Ruminococcaceae ↓ [34], [40] ^†^	Ruminococcus ↓ [34] ^†^ ↓ [40] (R. lactaris ↓ [37])
				Faecalibacterium ↓ [34] ^†^ (F. prusnitzii ↓ [37], [34] ^†^)
				Anaerofilum ↓ [34] ^†^
			Oscillospiraceae	Oscillibacter ↓ [34] ^†^
			Lachnospiraceae ↓ [40] ↑ [34] ^†^	Dorea ↑ [34] ^†^
				Rosaeburia (R. homnis ↓ [37], R. intesatinalis ↓ [37])
				Blautia ↑ [34] ^†^
				Coprococcus ↓ [37] (C. eutactus ↓ [37])
			Acidaminococcaceae	Megasphaera ↑ [34] ^†^
			Unclassified ↓ [34]	
	Negativicutes	Selenomonadales	Veillonellaceae ↑ [39]	
			Bifidobacteriaceae	Bifidobacterium ↓ [32,34] ↑ [44]
Verucomicrobia	Verucomicrobiae	Verucomicrobiales	Verucomicrobiaceae	Akkermansia ↓ [37]
Bacteroidetes ↓ [33]	Bacteroidia	Bacteroidales	Bacteroidaceae ↓ [33,39,40]	
			Porphyromonadaceae ↓ [39]	
			Prevotelaceae ↑ [38] ↓ [40]	
			Rikenellaceae	Alistipes (A. putredinis ↓ [37])
Proteobacteria ↑ [33]	γ-proteobacteria ↑ [33] *	Enterobacteriales	Enterobacteriaceae ↑ [40]	Klebsiella ↑ [37] (K. pneumonia ↑ [37])
				Enterobacter ↑ [44]
		Oceanospirillales	Halomonadaceae ↑ [40]	

* Mutlu EA, et al. Resluts on alcoholics with dysbiosis; ^†^ Leclerq S, et al and Bajaj JS, et al. Resluts on alcoholics with increased intestinsl permeability.

**Table 2 diseases-07-00058-t002:** Changes in intestinal microbiota associated with clinical studies on non-alcoholic fatty liver disease (NAFLD).

Phylum	Class	Order	Family	Genus
Firmicutes ↑ [90] ↓ [84,88,89]	Bacilli	Lactobacilales	Lactobacillaceae ↑ [89]	Lactobacillus ↑ [89,90,92]
			Streptococcaceae ↑ [91]	Streptococcus ↑ [92]
		Erysipelotrichales	Erysipelotrichaceae ↑ [91]	Anaerobacter ↑ [92]
	Clostridia ↓ [88]	Clostridiales ↓ [88]	Clostridiaceae	Anaerosporobacter ↓ [88]
				Anaerobacter ↑ [92]
				Clostridium (Cluster XI ↑ [92], C. leptum ↓ [111])
			Eubacteriaceae Ruminococcaceae ↓ [84,89,90,92]	Eubacterim (E. rectale ↓ [13]) Ruminococcus ↓ [89] ↑ [93] (R. obeum CA9:39 ↓ [13] R. obeum ↓ [13])
				Faecalibacterium ↓ [88,89] (F. prausnitzii ↓ [89])
				Oscillospira ↓ [93]
			Oscillospiraceae	Oscillibacter↓ [90,92]
			Lachnospiraceae ↑ [90,91] ↓ [84,89]	Dorea ↑ [90,93] ↓ [89]
				Roseburia ↑ [90] ↓ [89]
				Robinsoniella ↑ [90]
				Anaerostipes ↓ [89]
				Blautia ↑ [91] ↓ [89]
				Coprococcus ↓ [89]
			Incertae Sedis Family XI ↑ [92]	Peptoniphilus ↑ [93]
				Anaerococcus ↑ [93]
			Unclassified	Flavonifracter ↓ [92]
	Negativicutes	Selenomonadales	Veillonellaceae	Allisonella ↑ [88]
			Acidaminococcaceae	Phascolarctobacterium ↑ [94]
Lentisphaerae ↓ [92]				
Actinobacteria ↓ [84,94] ↑ [93]	Actinobacteria	Actinomycetales	Propionibacteriaceae	Propionibacterium (P. acnes ↑ [93])
Fusobacteria ↑ [91]				
Verrucomicrobia	Verrucomicrobiae	Verrucomicrobiales	Verrucomicrobiaceae	Akkermansia (A. muciniphila ↓ [100])
Bacteroidetes ↓ [73,93,111] ↑ [84,94]	Bacteroidia	Bacteroidales	Bacteroidaceae ↓ [89]	Bacteroides ↓ [89]
			Porphyromonadaceae ↑ [88] ↓ [89,90]	Parabacteroides ↑ [88] ↓ [89]
				Odoribacter ↓ [92]
			Prevotellaceae ↑ [84]	Prevotella ↑ [84,94] ↓ [91,92]
			Rikenellaceae ↓ [89,93]	Alistipes ↓ [89,92]
Proteobacteria ↑ [84,91]	γ-proteobacteria	Aeromonadales ↑ [88]	Succinivibrionaceae ↑ [88]	
		Enterobacteriales	Enterobacteriaceae ↑ [84,91,100]	Escherichia ↑ [84,91,92,103,116]
				Shigella ↑ [91]
	α-proteobacteria	Rhizobiales	Bradyrhizobiaceae	Bradyrhizobium ↑ [93]
		Rhodbacterales	Rhodbacteraceae	Rhodbacter ↑ [116]
	δ-proteobacteria	Desulfovibrionale	Desulfovibrionaceae	Biophila ↑ [116]

**Table 3 diseases-07-00058-t003:** Changes in intestinal microbiota associated with clinical studies on liver cirrhosis.

Phylum	Class	Order	Family	Genus (Speices)
Firmicutes ↑ [208] ^#^	Bacilli ↑ [38,187]	Bacillales	Stphylococcaceae	Staphylococcus ↑ [183] ^†^
		Lactobacilales	Lactobacillaceae	Lactobacillus (L. rhamnosus ↓84 L. fermentus ↓ [84], L. ruminis ↓ [37], L. antri ↑ [57], L. crispatus ↑ [57])
			Streptococcaceae ↑ [38,185,202]	Streptococcus ↑ [37,97] (S. thermophiles ↑ [37], S. infantarius ↑ [37])
			Enterococcaceae	Enterococcus ↑ [208] ^#^ (E. faecalis ↑ [182])
	Clostridia	Clostridiales	Clostridiaceae	Clostridium ↓ [182] ↑ [97], Clostridium XI ↑ [208] ^#^ Clostridium XIV ↓ [208] ^#^ [208] (C. methylpentosum ↓ [37], C. saccharolyricun [37] ↓)
			Eubacteriaceae	Eubacterium ↓ [97] (E. siraeum ↓ [37])
			Ruminococcaceae ↓ [37,188,214], [208] ^#^	Subdoligranulum ↓ [211]
				Faecalibacterium ↓ [37,182] (F. prausnitzii ↓ [184])
				Ruminococcus (R. sp 18P13 ↓ [37])
				Ruminococcacaeae bacterium ↓ [37]
			Lachnospiraceae ↓ [188,214], [208] ^#^	Dorea ↓ [211]
				Blautia ↓ [214]
				Butyrivibrio (B. crossotus ↓ [37])
	Negativicutes ↑ [185]	Selenomonadales	Veillonellaceae ↑ [38,187,188,204] ^#^	Veillonella ↑ [97], [208] ^#^ (V. dispar ↑ [185], V. paruvula [185] ↑)
			Acidaminococcaceae	Acidaminococcus ↑ [211]
				Phascolarctobacterium ↓ [37]
Actinobacteria ↑ [208] ^#^	Actinobacteriia	Bifidobacteriales	Bifidobacteriaceae	Bifidobacterium ↓ [180,184] ↑ [37] (B. catenulatum ↓ [185])
Fusobacteria [38,97] ↑	Fusobacteriia	Fusobacteriales	Fusobacteriaceae ↑ [38,188] ^#^	
Bacteroidetes ↓ [38,97,185,208] ^#^	Bacteroidia	Bacteroidales	Bacteroidaceae ↓ [38]	Bacteroides ↓ [97] (B. coprocola ↓ [37], B. uniformis ↓ [37])
			Prevotellaceae	Prevotella ↑ [97,208] ^#^↓ [37] (P. corpi ↓ [37])
				Paraprevotella ↓ [37] (P. xylaniphila ↓ [37])
			Rikenellaceae ↓ [208] ^#^	Alistipes ↓ [97] (A. shahil ↓ [37])
			Porphyromonadaceae	Barnecialla ↓ [37]
				Odoribacter ↓ [37] (O. splanchnicus ↓ [37])
				Parabacteroides (P. distasonis ↓ [37])
				Tannerella ↓ [37]
Proteobacteria ↑ [38,97,185,208] #	β-proteobacteria	Burkholderiales	Alcaligenaceae ↑ [188]	
			Burkholderiaceae	Burkholderia ↑ [214]
			Ralstoniaceae	Ralstonia ↑ [214]
	γ-proteobacteria ↑ [38,187]	Enterobacteriales	Enterobacteriaceae ↑ [38], [188] ^#^, [184,188,217]	Proteus ↑ [214]
				Escherichia ↑ (E. coli ↑ [183,187] †)

			Pasteurellaceae ↑ [38]	
	δ-proteobacteria	Desulfovibrionales	Desulfovibrionaceae	Biophila ↑ (B. wadsworthia ↓ [37])

^†^ Patients with minimal HE group; ^#^ Patients with HE group.

**Table 4 diseases-07-00058-t004:** Changes in intestinal microbiota associated with clinical studies on primary sclerosing cholangitis.

Phylum	Class	Order	Family	Genus
Firmicutes	Bacilli	Lactobacilales	Lactobacillaceae	Lactobacillus ↑ [241]
			Streptococcaceae	Streptococcus ↑ [249] ↑ # [249] (S. infantis ↑ [249] ↑ # [249] S. alactolyticus ↑ [252] ↑ # [252] S. equi ↑ [249] ↑] # [249], S. parasanguinis [252] ↑ # [252])
			Enterococcaceae	Enterococcus ↑ [241,249,252] (E. faecium ↑ [252], E.sp.NBRC107345 ↑ [252])
	Clostridia	Clostridiales	Clostridiaceae	Clostridium ↑ [249]
			Clostridiales II ↓ [239] ↓ # [239]	
			Ruminococcaceae	Ruminococcus ↑ # [246,252] (R. gnavus ↓ [249], R. obeum ↑ § [245])
				Faecalibacterium ↑ # [251] (F. prausnitzii ↓ [249])
				Dorea ↓ # [246]
				Roseburia ↑ # [251] ↓ # [246] ↓ [§267]
				Blautia ↑ § [245] ↓ # [246] (B. obeum ↑ § [245] ↓ [252], B. faecis ↑ § [245])
				Coprococcus ↓ [240,249] (C. catus ↓ [249])
			Lachnospiraceae ↑ § [248]	Anaerostipes (A. hardus ↓ [252])
				Lachnospira ↓ # [246]
	Negativicutes ↑ [187]	Selenomonadales	Veillonellaceae ↑ [38,187,204]	Veilonella ↑ [240,249,251] ↑ # [240,249] ↓ # [246]
			Acidominococcaceae	(V. dispar ↑ [249], ↑ # [249] V. paruvula ↑249], ↑ # [249] V.sp.3_1_44 ↑ [252]) Megasphaera ↑ [240] ↑ § [249] Phascolarctobacterium ↓ [240]
Actinobacteria	Actinobacteria	Actinomycetales	Micrococcaceae ↑ [249]	Rothia ↑ [249] ↑ # [249] (R. mucilaginosa ↑ [249])
		Coriobacteriales	Coriobacteriaceae	Adlercreutzia ↓ [249] (A. equolifaciens ↓ [249] ↓ # [249])
Fusobacteria	Fusobacteriia	Fusobacteriales	Fusobacteriaceae	Fusobacterium ↑ [241] ↑ # [246]
Bacteroidetes	Bacteroidia	Bacteroidales	Bacteroidaceae	Bacteroides ↓ § [249]
			Prevotellaceae	Prevotella ↓ § [249] (P. corpi ↓ [249], ↓ # [249])
			Porphydomonadaceae	Parabacteroides ↓ # [252] (P. distasonis ↓ [252])
			Barnesiellaceae ↑ [245]	
Proteobacteria	γ-proteobacteria	Enterobacteriales	Enterobacteriaceae	Escherichia ↑ § [249]
		Pasteurellales	Pasteurellaceae	Haemophilus ↑ [249]
		Aeromonadales	Succinivibrionaceae	Succinivibrio ↓ [240]
		Desulfovibrionales	Desulfovibrionaceae	Desulfovibrio ↓ [240]

# Bajer L et al. Resuls in patients with PSC (both with and without IBD.); # Iwasawa K, et al. Reslts in patients with PSC. # Rossen NG, et al. Reslts in patients with PSC compared with those with IBD and control. § Quraishi MN, et al. Results in patients with PSC compared with IBD and control. # Kummen M, et al. Results in patients with PSC compared with those with IBD and control.

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
