# Peer review of "Role of Gut Dysbiosis in Liver Diseases: What Have We Learned So Far?"

_diseases, 2019, doi:10.3390/diseases7040058_

Round 1

Reviewer 1 Report

acepted

Author Response

Dear respected reviewers,

Thank you very much for checking my review paper, giving reasonable comments and admitting your positive evaluations.

I checked it and revised the text as you pointed out. In addition, I corrected some typing mistakes.

Page 2, line 44; The text was revised as “At first, this chapter highlights the importance……” instead of “At first, I would like to strengthen the importance……” following the reviewer’s suggestion. I noticed some typing mistakes. I revised the text as follows.

Page 30, line 1103; The word “has” was changed to “have”.  

Abstract, line 29; The text should be written as “patients were associated” instead of “patients demonstrated that the patients were associated” Abstract, line 19; The phrase should be written as “more alcohol (alcohol hypothesis)” instead of “more alcohol”. Page 1, line 43, Introduction; The text was revised as “This review introduced” instead of “In this review, we introduced”. Page 3, line 97. 98; I corrected the phrase “The following sections introduce most of the outstanding researches in various clinical setting that suggest an important role of gut dysbiosis in the development of various liver diseases.” instead of “I would like to introduce most of the outstanding researches in various clinical setting that suggest an important role of gut dysbiosis in the development of various liver diseases.” This is to avoid the use od first person as the reviewer suggested. I found several typing mistakes in the present text. I marked all of them in blue characters. Please kindly correct all of them.

Reviewer 2 Report

The author well answered to my comments.

As minor issue, I only suggest to avoid the use of first person. 

In example, page 2, line 44: "At first, I would like to strengthen the importance ..." could be replaced by "At first, this chapter highlights the importance ..." and other similar in the text.

Author Response

Thank you very much for checking my review paper, giving reasonable comments and admitting your positive evaluations.

I checked it and revised the text as you pointed out. In addition, I corrected some typing mistakes.

Page 2, line 44; The text was revised as “At first, this chapter highlights the importance……” instead of “At first, I would like to strengthen the importance……” following the reviewer’s suggestion. I noticed some typing mistakes. I revised the text as follows.

Page 30, line 1103; The word “has” was changed to “have”.  

Abstract, line 29; The text should be written as “patients were associated” instead of “patients demonstrated that the patients were associated” Abstract, line 19; The phrase should be written as “more alcohol (alcohol hypothesis)” instead of “more alcohol”. Page 1, line 43, Introduction; The text was revised as “This review introduced” instead of “In this review, we introduced”. Page 3, line 97. 98; I corrected the phrase “The following sections introduce most of the outstanding researches in various clinical setting that suggest an important role of gut dysbiosis in the development of various liver diseases.” instead of “I would like to introduce most of the outstanding researches in various clinical setting that suggest an important role of gut dysbiosis in the development of various liver diseases.” This is to avoid the use od first person as the reviewer suggested. I found several typing mistakes in the present text. I marked all of them in blue characters. Please kindly correct all of them.

This manuscript is a resubmission of an earlier submission. The following is a list of the peer review reports and author responses from that submission.

Round 1

Reviewer 1 Report

Very interesting review but the quality of the figures is not so good, they are very bussy and dificult to follow.

Maybe divide them in 2 or3 more figures as a possible solution.

The review reads very well but sometimes it will be apreciated if the authors indicate if the data are in vivo; in vitro or from human studies.

Reviewer 2 Report

I think this manuscript is an important contribution to the literature. As a review, I believe the topic of this paper will provide service to the science community. Despite this importance and the considerable work done, I think this manuscript warrants some major improvements in order to be published.  

Chapter 1 needs important revision regarding the aim and content (title and sub-titles). This chapter focused on some bacterial products and in the same time with bile acids. I proposed:

to remove this chapter and improve the introductory section of the review or to improve this chapter by adding an introductory section to explain your aim and choices

As an example, the first sentence refers to patients with liver diseases, but translocation and bacterial components passage is also observed in "healthy" subjects. The roles of the 3 products (endotoxin, SCFAs and bile acids) are not very clear regarding normal or altered intestinal barrier. It would be better to explain gut-liver axis modifications (altered intestinal barrier,...) in liver diseases.

Chapter 2: The variations in gut microbiota according to the literature was never analyzed/discussed. All the chapter and especially the tables need to be more precise regarding the populations under assay (animals/humans; vs healthy subjects, vs other pathologies), the samples (bowel, mucosa, fecal, small intestine,....) and the sampling methods and at last analysis (cultural method/molecular biology; global analysis/specific detection or quantification)., interpretation (qualitative, quantitative, ratio,...). Some  of these important items have to be added in the tables. Regarding some controversy results, the authors may be more critical.

The last remark concerns some assertions: the gut dysbiosis provoks increased IP. Both items may be considered linked but currently I think that no one can claim the order of events

These major improvements should allow a meaningful conclusion.

Reviewer 3 Report

This cohomprensive review article focuses on gut dysbiosis related to several liver diseases.

Abstract is too long. Page 2, lines 47-49: it seems to be an introduction or the aim of the review, I suggest to add more. Provide higher resolution figure 2. Page 15: provide the legend for the table; the same for pages 21 and 24.